# PATHOLOGIES OF OUT-OF-DISTRIBUTION DETECTION

## ABSTRACT

There is a proliferation of out-of-distribution (OOD) detection methods in deep learning which aim to detect distribution shifts and improve model safety. These methods often rely on supervised learning to train models with in-distribution data and then use the models' predictive uncertainty or features to identify OOD points. In this paper, we critically re-examine this popular family of OOD detection procedures, revealing deep-seated pathologies. In contrast to prior work, we argue that these procedures are *fundamentally answering the wrong question* for OOD detection, with no easy fix. Uncertainty-based methods incorrectly conflate high uncertainty with being OOD, and feature-based methods incorrectly conflate far feature-space distance with being OOD. Moreover, there is no reason to expect a classifier trained only on in-distribution classes to be able to identify OOD points; for example, we should not necessarily expect a cat-dog classifier to be uncertain about the label of an airplane, which may share features with a cat that help distinguish cats from dogs, despite generally appearing nothing alike. We show how these pathologies manifest as irreducible errors in OOD detection and identify common settings where these methods are ineffective. Additionally, interventions to improve OOD detection such as feature-logit hybrid methods, scaling of model and data size, Bayesian (epistemic) uncertainty representation, and outlier exposure also fail to address the fundamental misspecification.

## 1 INTRODUCTION

In the real world, distribution shifts are the norm rather than the exception. We almost always have to deploy our predictive models on test points drawn from at least a somewhat different distribution than the training points: images acquired from different machines and hospitals, lane boundary detection in different cities, speech recognition with different accents (Amodei et al., 2016; Jung et al., 2021; Niu et al., 2016; Zhou et al., 2022; Koh et al., 2021). In order to make good predictions under these shifts, we need to build relevant *invariances* into our models, so that natural transformations such as rotations, translations, or even mild noise corruptions, do not significantly change the predictive distribution (Hendrycks and Dietterich, 2018; Mintun et al., 2021; Benton et al., 2020).

However, rather than *generalizing* to natural distribution shifts, it has become popular to *detect* out-of-distribution (OOD) data by training a supervised predictive model on in-distribution data, and then examining the model's uncertainty, logits, or features. A proliferation of works develop such procedures for detection improvements on known benchmarks (e.g., Hendrycks and Gimpel, 2016; Lee et al., 2018; Ren et al., 2021; Hendrycks et al., 2019a; Liang et al., 2017; Wang et al., 2021) or propose new and more challenging benchmarks (e.g., Hendrycks et al., 2019a; Yang et al., 2024; Wang et al., 2022; Bitterwolf et al., 2023; Yang et al., 2021).

While some prior work has considered limitations of OOD detection, the focus has been on issues with benchmarking (e.g., controlling for covariate shift versus semantic shift detection) (Yang et al., 2024), specific architectural features of generative models (e.g., normalizing flows with coupling layers) (Kirichenko et al., 2020), or minor method deficiencies that can be straightforwardly addressed (e.g., max-logit being more robust to many classes than max softmax) (Hendrycks et al., 2019a).

By contrast, we argue that *the whole premise of using supervised models trained on in-distribution data for out-of-distribution detection is fundamentally flawed* — a wholly misspecified enterprise, with no easy fix. First, the predictive distribution is over class labels, not whether an input comes from a different distribution: it is simply answering a different question. By making a predictive distribution better for detection, we could be removing invariances that would help with generalization. This effort is particularly self-defeating if the reason for wanting to do detection in the first place is

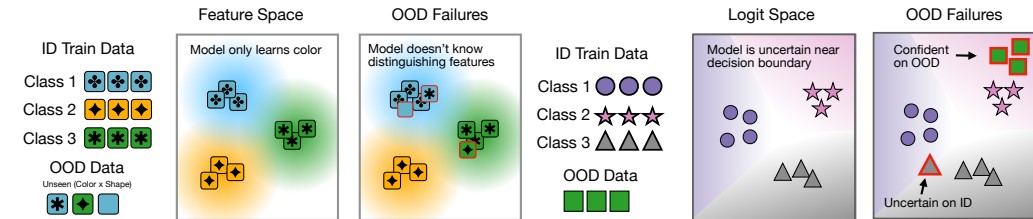

(a) Feature-based pathologies        (b) Logit-based pathologies

Figure 1: **There are irreducible errors when using supervised models for OOD detection because the problem is inherently misspecified.** Supervised models can only determine if an input leads to atypical representations or uncertain predictions, which is fundamentally different than determining if the input belongs to the training distribution.

to defer to a system that could provide reasonable generalization (e.g., thresholded classification). Moreover, there is often a focus on detecting *semantic shifts* (Yang et al., 2024), particularly new unseen classes, where the model in fact should not be expected to generally say anything reasonable. A car-truck classifier could be highly confident that a dog is a truck, if it uses similar features to distinguish trucks from cars. It is not that OOD detection is "fundamentally difficult", but rather that detection is being approached with methods that are fundamentally answering the wrong question — a dog on the whole may look much different than a truck, and should be distinguishable from trucks and cars, but not by training a supervised model only to differentiate between trucks and cars.

In this paper, we explain the fundamental conceptual limitations of popular supervised approaches for OOD detection, visualized in Figure 1, which we exemplify in several settings. We then consider the limitations — and, in some cases, additional pathologies — of standard interventions, such as epistemic (Bayesian) uncertainty representation and ensembling (Lakshminarayanan et al., 2017; Malinin et al., 2019; Tagasovska and Lopez-Paz, 2019; D'Angelo and Fortuin, 2021; Hendrycks et al., 2018; Pearce et al., 2021), introducing new unseen classes in the model predictive distributions (e.g., Fort et al., 2021), and outlier exposure (Hendrycks et al., 2018; Thulasidasan et al., 2021; Roy et al., 2022; Choi et al., 2023). We also critically examine other complementary procedures, such as generative models, which may appear to be more aligned with the question of OOD detection.

## 2 PRELIMINARIES

Let $f_\theta : \mathcal{X} \to \mathcal{Y}$ be a neural network with parameters $\theta$ which maps training data $X^{\mathrm{tr}} \sim p_\mathcal{X}(\cdot)$ to a class from $\mathcal{Y} = \{1, \ldots, K\}$. A model's decision function is derived from its predictive distribution $f_\theta(x) = \arg\max_{k \in \{1,\ldots,K\}} p_\theta(y = k|x)$. OOD detection methods, which leverage trained supervised models, propose a scoring function which for a test example $x^*$ assigns a scalar value $s(x^*, f_\theta, \mathcal{D}_{\mathrm{tr}})$ given a trained model $f_\theta$ and training data $\mathcal{D}_{\mathrm{tr}} = \{X_i^{\mathrm{tr}}, Y_i^{\mathrm{tr}}\}_{i=1}^N$. The score $s(x^*, f_\theta, \mathcal{D}_{\mathrm{tr}})$ is compared to a threshold value to determine whether $x^*$ will be detected as OOD or not. These methods are typically evaluated by computing AUROC (area under the receiver operating characteristic curve) scores on distribution shift benchmarks.

Two particularly common families of approaches have emerged for such OOD detection. If we view the model as a composition of transformations $p_\theta(y = c|x) = \mathrm{softmax}(c_\theta \circ e_\theta(x))_c$ where $e_\theta : \mathcal{X} \to \mathcal{F}$ is the penultimate layer feature extractor, and $c_\theta : \mathcal{F} \to \mathbb{R}^K$ is the classification layer outputting logits, then there are two natural signals to consider — features or logits.

**Feature-based approaches.** These methods compute the OOD score based on the features, typically from the penultimate layer. The most common approach is based on the squared *Mahalanobis Distance* (Maha) (Lee et al., 2018), where we fit a class-conditional Gaussian Mixture Model (GMM) to our features with $\mu_c = \frac{1}{N_c} \sum_{i:y_i=c} e(x_i), \Sigma = \frac{1}{N} \sum_{c=1}^K \sum_{i:y_i=c} (e(x_i) - \mu_c)(e(x_i) - \mu_c)^\top$. The Mahalanobis score is then computed from the negative of the squared Mahalanobis distance as

$$s_{\mathrm{Maha}}(x) = -\min_c \|\mu_c - e(x)\|_\Sigma^2 = -\min_c (x - \mu_c)\Sigma^{-1}(x - \mu_c)^\top,$$

Ren et al. (2021) extends this work and proposes *Relative Mahalanobis Distance*, which computes a likelihood ratio between the most likely class-conditional Gaussian and an unconditional Gaussian fit

over all train data with $\mu_{\text{train}} = \frac{1}{N} \sum_i e(x_i)$ and $\Sigma_{\text{train}} = \frac{1}{N} \sum_i \big( e(x_i) - \mu_{\text{train}} \big) \big( e(x_i) - \mu_{\text{train}} \big)^\top$.
The Relative Mahalanobis score is $s_{\text{RelMaha}}(x) = -\min_c \|\mu_c - e(x)\|_\Sigma^2 - \|\mu_{\text{train}} - e(x)\|_{\Sigma_{\text{train}}}^2$.
Many other feature-based approaches have also been proposed (Sun et al., 2022; Tack et al., 2020; Sehwag et al., 2021).

**Logit-based approaches.** These methods operate on the logits of a trained supervised model. The most common approach is *Maximum Softmax Probability* (MSP) (Hendrycks and Gimpel, 2016) $s_{\text{msp}}(x) = \max_c p_\theta(y = c|x)$. Other popular approaches within this family include the entropy of the predictive distribution $p_\theta(y|x)$ (Ren et al., 2019), value of the max logit (Jung et al., 2021) and the energy score (Liu et al., 2020).

Despite a proliferation of methods, simple approaches such as MSP tend to provide state-of-the-art results, even on the more sophisticated benchmarks (Hendrycks et al., 2019a; Yang et al., 2024). For example, in Table 1 of Yang et al. (2024), it is observed that *"the results confirm that MSP still outperforms all modern methods"*. These methods are thus a natural choice to exemplify the broad conceptual issues with OOD detection, since they provide simple, popular, and still highly competitive approaches.

**OOD detection task.** These OOD detection methods can be used on various types of OOD detection. Much work has focused on using supervised models to identify points where the model does not have any chance of a correct label, often referred to as *semantic shift*, as opposed to label-preserving *covariate shift*. This type of semantic shift detection is often further categorized into *near OOD*, where the points are similar, and *far OOD* for more distinct inputs.

## 3 RELATED WORK

While anomaly and outlier detection has been studied for many decades in statistics, the related but distinct area of *out-of-distribution* (OOD) detection in deep learning is surprisingly new. Amodei et al. (2016) provides a call to action to build methods that are robust to distribution shifts. Shortly after, Hendrycks and Gimpel (2016) proposed using softmax uncertainty as a simple baseline to detect out-of-distribution (OOD) points. A proliferation of methods followed, using the logits, features, or uncertainty of a supervised model trained on in-distribution data to detect out-of-distribution points, achieving better results on benchmark detection tasks (Lee et al., 2018; Liang et al., 2017; Wang et al., 2022; Sun et al., 2022). Other work has focused on introducing new benchmarks with higher resolution images, or test detection, more specifically under *semantic shift* (e.g., new unseen classes) versus covariate shift (label-preserving transformations) (Yang et al., 2024; Bitterwolf et al., 2023; Huang and Li, 2021). There are also many interventions for boosting performance, including Bayesian uncertainty representation (Lakshminarayanan et al., 2017; Malinin et al., 2019; Tagasovska and Lopez-Paz, 2019; D'Angelo and Fortuin, 2021; Rudner et al., 2022), confidence minimization and outlier exposure (Hendrycks et al., 2018; Papadopoulos et al., 2021; Thulasidasan et al., 2021), and pre-training (Fort et al., 2021; Tran et al., 2022; Hendrycks et al., 2019b).

While there are several works critical in some way of OOD detection, our focus is significantly different. Critiques tend to be targeted at modifications to existing measures (e.g., max-logit has fewer false positives than MSP) (Hendrycks et al., 2019a), improving the benchmark data (e.g., higher resolution data, data with many classes, and more cleanly separating semantic shift from covariate shift) (Hendrycks and Dietterich, 2019; Yang et al., 2024), specific architectural properties of generative models (e.g., coupling layers in normalizing flows) (Kirichenko et al., 2020), or note that detection might need to be more tailored to specific shifts (Tajwar et al., 2021; Farquhar and Gal, 2022). By contrast, we examine whether the predominant approach of training supervised models on in-distribution data for OOD detection is *fundamentally misspecified*, answering a different question than "is this point out-of-distribution?" We conceptually elucidate significant pathologies of both feature and logit-based approaches to OOD detection, and then exemplify these pathologies. We also show that interventions such as Bayesian (epistemic) uncertainty have their own pathological behavior, despite being considered the principled approach to OOD detection in prior work. We further show that other interventions, such as confidence minimization, can introduce a trade-off between detection and generalization. Moreover, we consider whether generative models are more directly answering the question "is this point from a different distribution?" and also evaluate the deficiencies of such approaches. We also directly contrast these approaches, and simple statistical baselines, with the supervised methods. Viewing these issues through the lens of misspecification, we finally consider interventions that can help reduce misspecification to improve detection performance.

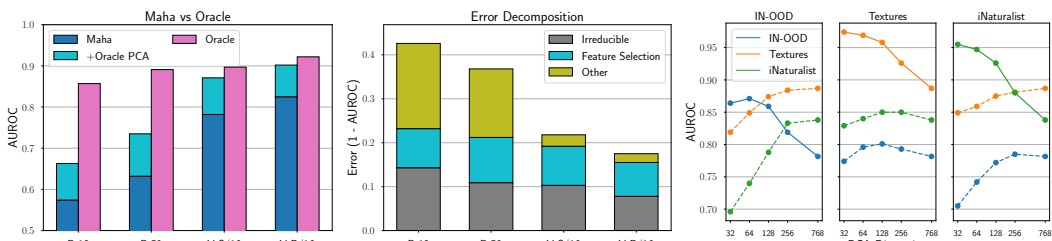

Figure 2: **Feature-based methods have two key failure modes: indistinguishable features and irrelevant features. (Left):** Oracle classifier achieves only around 90% AUROC on ImageNet vs ImageNet-OOD, indicating some OOD inputs have *indistinguishable features*. Oracle PCA projection improves Mahalanobis AUROC, showing many features are *irrelevant for OOD detection*. **(Middle):** Error decomposition into irreducible, suboptimal feature selection, and other components. **(Right):** Top discriminating features are OOD dataset-specific. PCA on ID and $OOD_A$ improves AUROC for $OOD_A$ (solid line, name in title) but decreases it for $OOD_B$ (dashed lines) for ViT-S/16 features.

## 4 OOD Detection Methods Answer the Wrong Questions

Many OOD detection methods rely on the features or logits from supervised models that are only exposed to in-distribution data. Even though these approaches are sometimes able to achieve reasonable results on OOD detection benchmarks, they fundamentally answer the wrong question: instead of determining whether an input belongs to the training distribution or some different distribution, they instead ask if the input leads to atypical model representations or unconfident predictions. In this section, we explore the concrete instances where the answers to these two questions differ, and we demonstrate that feature and logit-based OOD detection methods have irreducible errors as a result.

### 4.1 Feature-Based Methods

Feature-based methods typically use distance metrics to measure how close the features of the test input are to the features of the train inputs, answering the question "does this input lead to features that are far from the features seen during training?". These methods have two fundamental failure modes: 1) the learned features do not sufficiently discriminate between OOD and ID inputs, and 2) the optimal distance metric depends on the OOD data, forcing these methods to use suboptimal, heuristic-based distance metrics given only access to ID data. In particular, only the distance information along a small number of feature dimensions is useful for OOD detection, but it is impossible to infer these most discriminating features from irrelevant features without access to OOD data.

**OOD features can be indistinguishable from ID features.** While OOD inputs generally have unique characteristics that distinguish them from ID data, a supervised model may not be incentivized to learn these features if they are unhelpful for ID classification. If the OOD and ID features are indistinguishable, then no feature-based methods can perform well. This failure mode may have especially significant impacts for near OOD detection where fine-grained features are required.

To demonstrate this lack of separability between ID and OOD features, we study four different models trained on ImageNet-1k: ResNet-18, ResNet-50, ViT-S/16, and ViT-B/16, with the OOD datasets of ImageNet-OOD (Yang et al., 2024), Textures (Cimpoi et al., 2014), and iNaturalist (Van Horn et al., 2018). For each setting, we train an Oracle, a binary linear classifier, to differentiate between ID features and OOD features and report its performance on held-out ID and OOD features. This Oracle serves as a proxy for the best possible performance of any feature-based OOD detection method since it is directly trained on both ID and OOD features, unlike any realistic methods. We see in Figure 2 (left) that even with ground-truth OOD information, the Oracle is unable to clearly disambiguate between ID and OOD examples on challenging OOD datasets, obtaining AUROCs as low as 0.86. For each model, $(1 - $ Oracle AUROC$)$ represents an irreducible error: no feature-based method can correctly detect these OOD inputs that have indistinguishable features from ID.

**Irrelevant features hurt performance and are impossible to fully identify.** Even if the model has learned features that discriminate between OOD and ID data, it is generally impossible to identify which features discriminate between OOD and ID data and which features are irrelevant without access to OOD data. As a result of the underspecification of OOD data at train-time, feature-based

methods must resort to suboptimal distance metrics to compare OOD features from ID features that do not sufficiently up-weight discriminating features or down-weight irrelevant features.

We illustrate this failure mode with the features from ResNet-18, ResNet-50, ViT-S/16, and ViT-B/16 trained on ImageNet-1k, and the Mahalanobis (Maha) method, which uses a distance metric defined by the empirical covariance matrix $\Sigma$ of ID features. We compare the performance of Maha before and after an Oracle PCA projection that preserves only the most discriminating dimensions between OOD and ID features, computed by performing PCA on both ID and OOD features and using the number of PCA components among $\{32, 64, 128, 256\}$ that maximizes the resulting Maha AUROC. In Figure 2 (left), we show the addition of an Oracle PCA projection significantly improves Maha performance on all models by an average of over 10 percentage points. Moreover, we see in Figure 2 (middle) that performing this PCA projection accounts for *nearly all* of the reducible error of Maha for the ViT models. In other words, for ViTs, the gap between Maha and the best possible performance 1) is almost entirely explained by the use of irrelevant features in the distance computation, and 2) requires information unavailable to any feature-based method. While methods such as Relative Mahalanobis and ViM (Ren et al., 2021; Wang et al., 2022) use PCA projections or related ideas to attempt to reduce the impact of irrelevant features, they can only use feature covariances computed on ID data alone, and thus do not address this fundamental limitation as we show in Appendix A.1.

In Figure 2 (right), we show that the Oracle PCA projection is highly specific to the particular OOD dataset we wish to detect and does not transfer between OOD datasets. For example, as demonstrated in the first panel, using the top 32 PCA components computed on IN and IN-OOD improves Maha AUROC in detecting IN-OOD but significantly degrades the AUROC for detecting Textures and iNaturalist, using features from ViT-S/16. This result shows that, as long as the OOD dataset is not specified at training time, removing the influence of irrelevant features is impossible for any feature-based method, presenting another fundamental bottleneck to its detection performance.

**Visual demonstrations.** We visualize clear examples of failure modes for feature-based methods in Appendix A.1. To demonstrate feature overlap, we train a ResNet-18 on a subset of CIFAR-10 classes: airplane, cats, and trucks. We then use this trained model to detect OOD images of dogs. We see in Figure A.1 (left) that the feature space between cats and dog have very high overlap, since the model did not learn the features necessary to distinguish between these two classes. This pathology is reflected in the poor performance of feature-based methods such as Mahalanobis distance, which only achieves an AUROC of 0.537 and is barely better than random chance. Furthermore, these failures also occur in larger models trained on diverse datasets. Even when using a ResNet-50 trained on ImageNet-1K, Figure A.1 (right) demonstrates that feature-based methods like Mahalanobis distance fail to correctly differentiate ID from OOD examples and assign low distances to OOD inputs.

## 4.2 LOGIT-BASED METHODS

Due to the many pathologies of feature-based OOD detection methods, it may be tempting to instead focus on logit-based methods, which gauge a model's uncertainty over an input's predicted labels via its logits. However, the previous limitations are still applicable. For instance, in the scenario where OOD and ID features overlap, logit-based methods would also fail to detect OOD inputs since the logits are a function of the penultimate-layer features. Furthermore, logit-based methods have their own suite of failure modes which arise from the conflation of *label uncertainty*, the uncertainty over the correct ID label, with *OOD uncertainty*, the uncertainty over whether the sample is ID or OOD. Logit-based methods heuristically assume that higher label uncertainty is equivalent to higher OOD uncertainty, but these are fundamentally different quantities. As a result, there are two distinct failure modes where logit-based methods make the incorrect prediction: instances where ID data naturally has high label uncertainty, and instances where OOD data has low label uncertainty.

**ID examples often have high uncertainty.** To show the misalignment between label and OOD uncertainty, we demonstrate instances where models predict high label uncertainty over in-distribution samples. One example of this failure mode can be found in ImageNet-1K, where it is known that many of the images within the dataset contain concepts from multiple classes (Stock and Cisse, 2018; Shankar et al., 2020). We would expect these multi-label images to have high label uncertainty since there may be multiple correct answers. For our experiments, we used the human annotations from Beyer et al. (2020) as the ground truth for the number of labels corresponding to each image.

We explore the behaviors of ResNet-18, ResNet-50, ViT-S/16, ViT-B/16, all trained on ImageNet-1k, as well as ViT-G/14 DINOv2 pretrained on internet-scale data, for uncertainty-based OOD detection.

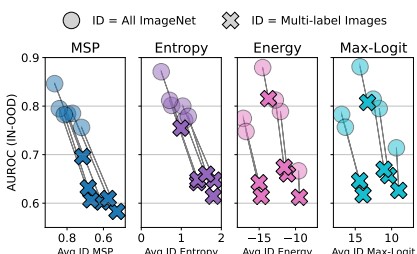
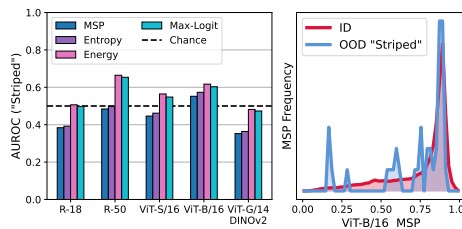

(a) Label uncertainty is high but input is ID      (b) Label uncertainty is low but input is OOD

Figure 3: **Logit-based methods incorrectly conflate label uncertainty with OOD uncertainty**. **(Left):** ID images with multiple correct labels should have high label uncertainty. Each connected line shows the decrease in OOD detection for models listed in the right panel when focusing on this subset of high-uncertainty ID data. **(Right):** All methods assign low uncertainty to the OOD class 'Striped' from Textures and perform similarly to random chance.

When we apply uncertainty-based metrics to these samples where multiple labels may apply, we find in Figure 3 that the average uncertainty of these multi-label images, denoted with Os, is significantly higher than corresponding in-distribution samples, denoted with the connected Xs, across a variety of methods. However, these images are clearly ID, since they are sampled from the same distribution that the model was trained on. Furthermore, we can also see that logit-based methods are not able to distinguish between ID inputs with high natural label uncertainty and OOD inputs; for example, the AUROC for multi-label images (ID with high label uncertainty) vs ImageNet-OOD is only around 0.6. These results reveal that uncertainty-based methods are insufficient for OOD detection.

**OOD examples often have low uncertainty.** In Figure 3, we consistently find that logit-based approaches are unable to distinguish between ID and the "Striped" class from Textures across many settings. Furthermore, in Table A.1, we benchmark 14 different models including ResNets, ViTs, and ConvNext V2 in the setting where ImageNet-1K is ID. We record the FPR@95, which indicates how many OOD examples are incorrectly classified as ID due to their low uncertainty (false positive), at a threshold where 95% of ID examples are correctly classified. For logit-based methods such as MSP, max-logit, energy score, and entropy, the average FPR@95 across all settings is over 60%; thus, a majority of OOD examples are misclassified due to their low uncertainty.

We provide visual examples of these failure modes of uncertainty-based methods in Appendix A.2, where the predictive uncertainties of ID inputs are indistinguishable from the uncertainties of OOD inputs. In Figure A.3, we note how the uncertainties of an ID and OOD class entirely overlap for a LeNet-5 trained on a subset of CIFAR-10. We also visualize the feature space of a ResNet-50 trained on ImageNet-1k in Figure A.4 and find that the OOD class is often far from the decision boundary and has high model confidence, even though the examples are not from the input distribution.

Our experiments demonstrate that the difference between label uncertainty and OOD uncertainty, although easy to miss, is a fundamental limitation of logit-based OOD detection methods. This inherit misalignment of goals means no logit-based methods can overcome this pathology.

## 5 BUT WHAT ABOUT …?

Given the prevalence of failure modes when using only feature or logit-based OOD methods, numerous strategies have been proposed to enhance OOD performance. In this section, we examine popular interventions such as combining feature and logit-based approaches, pre-training on larger datasets, modeling epistemic uncertainty, and exposing the model to outliers. For these methods, we analyze their limitations, and demonstrate how they fail to address the fundamental pathologies outlined in Section 4. We also address the limitations of explicitly including an OOD class during training and using unsupervised generative models.

### 5.1 SCALING MODEL AND DATA SIZE

Increasing model size and pre-training on large datasets have been shown to reliably improve OOD detection benchmarks as models tend to learn more diverse and higher-quality features (Fort et al., 2021; Dehghani et al., 2023; Miyai et al., 2023). When models see more diverse data and as the model capacity increases, they can learn more features that help distinguish OOD and ID data.

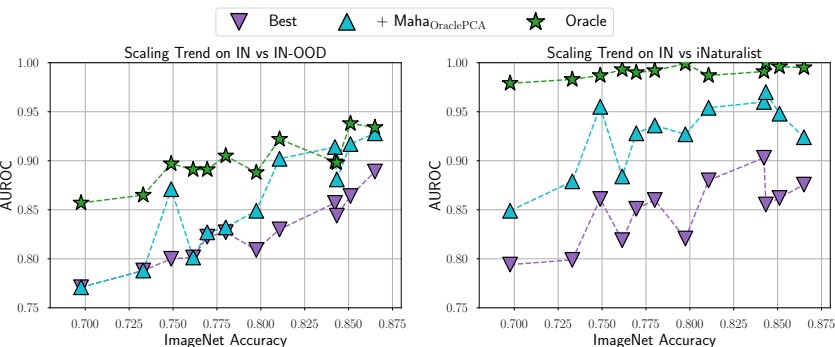

Figure 4: **Scaling model size and training data does not address the fundamental limitations.**
Scaling from ResNet-18 trained on ImageNet (left-most point) to ViT-G/14 DINOv2 pre-trained
on internet scale data (right-most), the Oracle AUROC still shows significant irreducible error for
IN-OOD. Furthermore, a large fraction of the gap between the best performing method (described in
text) and the Oracle can also be recovered by selecting features through Oracle PCA, indicating that
the influence of irrelevant features is not addressed by scale.

However, as we show in Figure 4, scaling alone does not fully address the limitations of OOD
detection methods. We benchmark twelve different models of varying sizes and pretraining methods,
enumerated in Appendix B.3. First, in challenging near-OOD detection problems such as ImageNet
vs. ImageNet-OOD, models learn additional discriminating features between ID and OOD data at an
extremely slow rate, such that even the largest ViT-G/14 DINOv2 still has over $5\%$ irreducible error
due to indistinguishable features. As we have argued in Section 4, this error can not be decreased
regardless of what OOD detection method we use. Indeed, the AUROC achieved by the best method
(Best) among Maha, Rel Maha, MSP, Max Logit, Energy, and ViM is consistently below the Oracle,
a binary classifier trained on ID and OOD features, by a wide margin. Second, while the error due
to indistinguishable features may be decreased (slowly) with scale and can become negligible on
far-OOD detection problems such as ImageNet vs iNaturalist, there is still a large gap between the
best existing method and the Oracle as we scale the model. Much of this gap can be recovered by the
gain from optimally selecting features for Maha, represented by $+\mathrm{Maha}_{\mathrm{OraclePCA}}$ (the gain is zero
if Best already outperforms $\mathrm{Maha}_{\mathrm{OraclePCA}}$), suggesting that while scaling allows the model to learn
features which almost perfectly discriminate ID and OOD data, the presence of irrelevant features
continues limit the performance. We provide additional empirical results in Appendix A.4 which
demonstrate the scaling behaviors of logit-based methods using 54 models over nine architectures
and six pre-training setups. These results demonstrate the fundamental limitations of existing OOD
detection methods even with increasing model and data size.

## 5.2 Combining Feature and Logit-Based Methods

Hybrid approaches which combine model features and logits have been proposed for OOD detection
(Sun et al., 2021; Wang et al., 2022), and methods like Virtual-logit Matching (ViM) (Wang et al.,
2022) have achieved state-of-the-art results for certain OOD benchmarks. To understand the success
of these methods, we test a simple hybrid method which sums the normalized scores of a feature-based
method (Mahalanobis) with a logit-based method (MSP), to which we refer as "Hybrid-Add". We
find in Appendix A.3 that for some models, "Hybrid-Add" improves OOD detection compared to
using MSP or Mahalanobis alone on the Textures dataset, indicating feature and logit-based methods
can have distinct failure modes.

However, hybrid methods do not address the fundamental pathologies caused by the model mis-
specification. In the many cases where ID and OOD features are indistinguishable, as described in
Section 4.1, hybrid methods are equally unable to differentiate OOD examples because both features
and logits will overlap. Furthermore, the usefulness of hybrid methods is largely dataset-dependent.
For instance, we find that ViM usually outperforms both MSP and Mahalanobis for OOD detection
on Textures (Figure A.6) but does not offer a consistent advantage on IN-OOD (Figure 5). Our simple
"hybrid-add" method does not offer a clear and consistent advantage over MSP or Mahalanobis on
either IN-OOD or iNaturalist. In practice, we find that feature and logit-based methods do not always
share distinct failure modes, and so hybrid methods may not be beneficial.

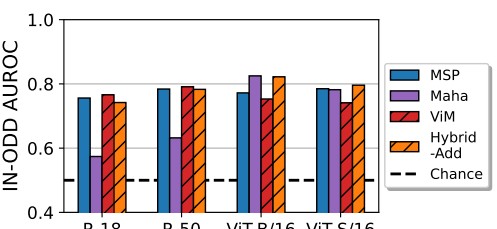 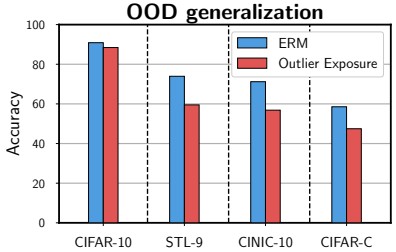

Figure 5: Hybrid OOD methods may not be beneficial. We compare two hybrid methods (ViM, Hybrid-Add) against MSP and Maha.

Figure 6: Training a ResNet-18 with outlier exposure hurts OOD generalization for covariate shifts compared to standard training.

## 5.3 EXPOSING TO OUTLIERS

Another popular approach to improve OOD detection is outlier exposure, which incorporates OOD examples when training the model (Hendrycks et al., 2018; Choi et al., 2023). In this setting, we explicitly optimize the model to have high uncertainty on the outlier dataset:

$$\mathcal{L}_{\text{CE}} + \alpha\mathcal{L}_{\text{OE}} = \mathbb{E}_{(x,y)\sim\mathcal{D}_{\text{in}}}\ell_{\text{CE}}(f(x),y) + \alpha\mathbb{E}_{x'\sim\mathcal{D}_{\text{out}}}\ell_{\text{CE}}(f(x'),y_u)$$

where $y_u$ is uniform distribution over all $K$ classes. Outlier exposure relies on the diverse dataset $\mathcal{D}_{\text{out}}$ in order to encourage the model to generally have high predictive uncertainty away from the training data and improve detection with predictive-space methods like MSP. However, even if the model is exposed to OOD data during training, the final model is still misspecified because it only contains ID classes as possible categorizations. As previously discussed in Section 4.1, OOD datasets are quite diverse, and the features necessary to distinguish ID from one OOD dataset often do not generalize to other types of OOD.

Furthermore, *outlier exposure may significantly hurt OOD generalization* because the model is explicitly trained to have high label uncertainty over a large set of inputs; this degradation in performance is especially problematic because OOD generalization is essential for model robustness and reliability. To demonstrate this behavior, we compare two ResNet-18 models trained on CIFAR-10, one with the standard training regime and the other with outlier exposure using TIN-597 as $D_{out}$ following Zhang et al. (2023) (see Appendix B.1 for setup details).

In Appendix A.6, we show that outlier exposure does improve OOD detection for most of the semantic shift OOD benchmarks. However, outlier exposure does not improve performance on MNIST, likely because this dataset differs significantly from $D_{out}$ and other natural image benchmarks. This decreased performance highlights the sensitivity of outlier exposure to the choice of OOD data and reiterates that the features which distinguish ID and OOD are not consistent across diverse OOD datasets. Furthermore, we find that while the ID accuracy of the outlier exposed model is negatively impacted, the impacts of outlier exposure on OOD generalization is significantly worse. In Figure 6, on inputs with covariate shifts, outlier exposure hurts the model's accuracy by over 10% across all of our benchmarked datasets. Thus, by explicitly encouraging high uncertainty on the diverse outlier dataset, we sacrifice the generalizability of our model.

## 5.4 MODELING EPISTEMIC UNCERTAINTY

Predictive uncertainty can be separated into *aleatoric uncertainty*, which is considered irreducible and stems from inherent data variability, and *epistemic uncertainty*, which is uncertainty over which solution is correct given the limited data. It has been posited that focusing on epistemic uncertainty for predictive models is the principled approach to OOD detection because the uncertainty increases as we move away from the data, and indeed there is a proliferation of methods approximating epistemic uncertainty for improved performance on OOD detection benchmarks (e.g., Band et al., 2021; D'Angelo and Fortuin, 2021; Lakshminarayanan et al., 2017; Malinin et al., 2019; Rudner et al., 2022; Tagasovska and Lopez-Paz, 2019; Tran et al., 2022).

Epistemic uncertainty is typically represented through a distribution over the model parameters. For a model $f$ with stochastic parameters $\Theta$, distributed according to $q(\theta)$, we can express the model's predictive uncertainty as a combination of aleatotric and epistemic uncertainty,

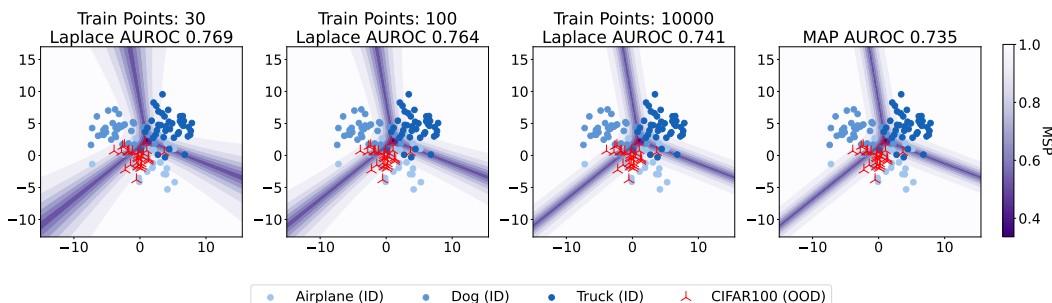

Figure 7: **Epistemic uncertainty becomes less useful for OOD detection as more in-distribution data is observed.** We consider three CIFAR-10 classes as ID, and CIFAR-100 as OOD. We use a ResNet-18 with a last-layer Laplace approximation to measure epistemic uncertainty. As we increase the in-distribution training examples, the posterior collapses and epistemic uncertainty diminishes.

$$\underbrace{\mathcal{H}\left(\mathbb{E}_{q_{\Theta}}[p(y \mid \mathbf{x}, \Theta)]\right)}_{\text{Total Uncertainty}} = \underbrace{\mathbb{E}_{q_{\Theta}}[\mathcal{H}(p(y \mid \mathbf{x}, \Theta))]}_{\text{Aleatoric Uncertainty}} + \underbrace{\mathcal{I}(Y; \Theta)}_{\text{Epistemic Uncertainty}}, \tag{1}$$

where $\mathcal{H}(\cdot)$ is the entropy functional and $\mathcal{I}(Y; \Theta)$ is the mutual information. The predictive distribution, through the rules of probability, is then $p(y = c | \mathbf{x}, \mathcal{D}) = \int \text{softmax}(f_{\theta}(\mathbf{x}))_c \cdot p(\theta | \mathcal{D}) d\theta$. We note that *deep ensembling* procedures (Lakshminarayanan et al., 2017), particularly popular for OOD detection, are a prominent example of epistemic uncertainty representation; by marginalizing over modes in a posterior they often provide a relatively accurate representation of the posterior predictive distribution (Wilson and Izmailov, 2020; Izmailov et al., 2021b).

However, as we have previously discussed, the predictive uncertainty is not over whether a point is OOD, but rather over class labels. Epistemic uncertainty does not address this fundamental misspecification. For a clear demonstration of the conceptual difference between epistemic uncertainty and OOD detection, consider how epistemic uncertainty changes as a function of data size. In the infinite ID-data limit, the epistemic uncertainty of a model approaches zero and the model becomes extremely confident in its parameters. If measuring epistemic uncertainty were the correct approach to OOD detection, then having such low epistemic uncertainty implies that OOD points do not exist in this setting. Therefore, because perfectly capturing epistemic uncertainty is not enough to solve OOD detection, they must answer fundamentally different questions. In fact, as the model sees more in-distribution data during training, its ability to detect OOD inputs may become worse! Given the growing availability of large datasets, this behavior becomes increasingly problematic.

To illustrate this phenomenon, we consider a last-layer Bayesian approximation (Kristiadi et al., 2020) and train a linear layer $f_{\theta}$ over features extracted from a ResNet-18 trained on IN-1K to classify three classes: airplane, dog, and truck. We place a prior over parameters $\theta$, and we approximate the predictive distribution through a Laplace approximation that uses a Gaussian distribution to approximate the posterior distribution of the model parameters, allowing for the estimation of epistemic uncertainty (MacKay, 2003). In Figure 7, we visualize the learned decision boundaries by applying PCA to reduce the three-dimensional logit space to two dimensions and plot the the maximum softmax probabilities over this projection. As the size of the training data increases, the posterior noticeably contracts, and the performance of the Laplace model decreases to approach the performance of the deterministic model with maximum a posteriori (MAP) parameter estimates.

In other work, Izmailov et al. (2021a) studies pathologies in Bayesian methods for OOD generalization (rather than detection), and D'Angelo and Henning (2021) note the sensitivity to prior specification in using BNNs for OOD detection.

## 5.5 INTRODUCING AN UNSEEN CLASS

Since standard classification models trained over $K$ classes are fundamentally misspecified for the task of detecting OOD classes in both feature-space and logit-space, it may be tempting to correct the specification problem by adding a $(K + 1)$-th class corresponding to the OOD category (Thulasidasan et al., 2020). During training, we can then expose models to OOD examples, and use this additional class for OOD detection on test samples.

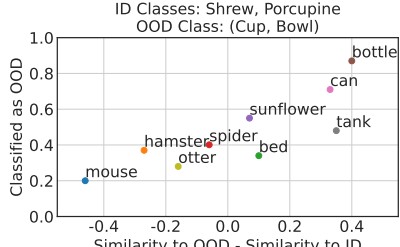

Figure 8: Adding an OOD class is only effective if the train OOD examples are similar to test OOD examples.

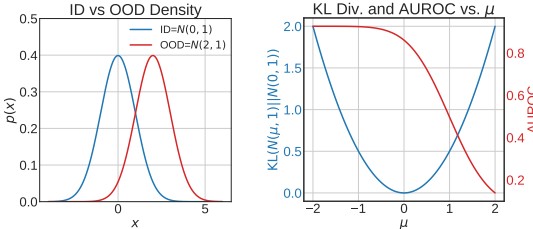

Figure 9: Generative models are not optimal OOD detectors: for generative model $\mathcal{N}(\mu, 1)$, the optimal $\mu$ is $0$ to model the ID data but $-\infty$ for OOD detection.

However, we find this method is only effective when the examples that the model is exposed during training are very similar to the OOD examples during test-time, which is often unrealistic. To demonstrate, we train a ResNet-18 model on two CIFAR-100 classes: keyboard and porcupine, and use samples from cup and skyscraper for the OOD class. We then measure the performance of OOD detection over the remaining CIFAR-100 classes. In Figure 8, we use BERT embeddings (Devlin et al., 2018) to compute the cosine similarity of the test-time OOD classes to the train-time ID and OOD classes. We see that the OOD class was effective in capturing test-time examples of bottle and can, since they are similar to the train OOD examples. However, the model is unable to accurately categorize examples like hamster and mouse, which are more closely related to ID classes.

### 5.6 USING GENERATIVE MODELS

Unlike previously mentioned methods utilize a supervised classification problem to make predictions, unsupervised generative models trained on the in-distribution dataset attempt to directly answer the question of how likely it is that sample $x$ belongs to the training distribution. Generative models, therefore, may appear to be a principled and natural solution to OOD detection.

However, better generative models are not always better OOD detectors. Since $p(x)$ answers a fundamentally different question than $p(\text{OOD}|x)$, there is generally a conflict between creating a better model for $p(x)$ and the ability to use the likelihood from that model to detect OOD points. We illustrate this phenomenon with a simple 1D example in Figure 9, where the ID data is drawn from $\mathcal{N}(0, 1)$ and the OOD data is drawn from $\mathcal{N}(2, 1)$. Suppose we model the ID data with $x \sim p_\mu(x) = \mathcal{N}(x|\mu, 1)$, where $\mu$ is the parameter of our model. Choosing $\mu = 0$ will exactly model the true distribution and achieve the highest likelihood. However, as shown in Figure 9 (right), the optimal choice of $\mu$ for OOD detection is $-\infty$, achieving a maximum AUROC but infinite KL divergence from the true ID distribution. We further demonstrate this misalignment between likelihood on the ID data vs OOD detection in Appendix A.5, discussing additional limitations and illustrating the failures of generative approaches such as diffusion models for OOD detection.

## 6 DISCUSSION

Fundamentally, we have shown that popular OOD detection procedures, both supervised and generative, are often answering a different question than *is this unlabeled point from a different distribution?* Moreover, interventions like outlier exposure hurt the ability for a model to generalize on covariate shifts which begs the question: *why are we doing OOD detection in the first place?* If it is really to detect OOD points, then the procedures we are using are severely misspecified and have fundamental limitations. If the goal of OOD detection is to be able to make more reasonable predictions under covariate shifts (e.g., by deferring examples to another model through confidence thresholding), which is arguably a more typical real-world use case than semantic shift, the interventions for detection can be actively harmful.

Going forward, it will be important to identify real-world problems where one of generalization or detection is the focus — and in the cases where the ultimate objective really is detection, we should build approaches specifically designed to answer that question.

## ETHICS STATEMENT

The authors have read and they acknowledge the ICLR Code of Ethics. The authors will strictly adhere to the ICLR Code of Ethics.

## REPRODUCIBILITY STATEMENT

Key details to reproduce experiments are provided in the main text and appendix. In addition, we intend to release the code publicly to reproduce all data and figures presented in the paper.

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

TABLE OF CONTENTS

## A  ADDITIONAL EMPIRICAL RESULTS

### A.1  FEATURE-BASED METHODS

We provide visualizations of empirical examples of feature-based failures in Figure A.1, and we demonstrate that advanced methods like relative Mahalanobis and ViM are subject to the same failure modes in Figure A.2.

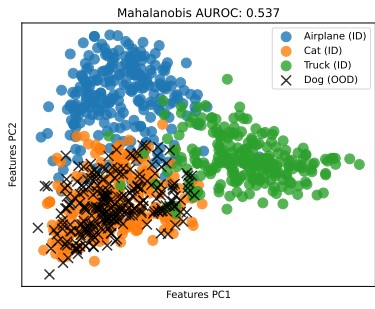
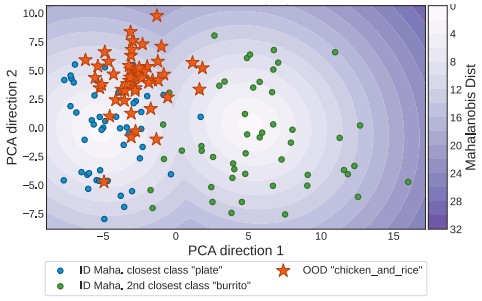

(a) ResNet-18 on CIFAR-10          (b) ResNet-50 on ImageNet-1k

Figure A.1: Visualizations of failure modes for feature-based OOD detection. **(Left):** We train a ResNet-18 on a subset of CIFAR-10, and find that the feature space between an ID class and OOD class have significant overlap. **(Right):** Feature-based methods also fail for larger models like ResNet-50 trained on ImageNet 1K, where OOD classes have low Mahalanobis distance.

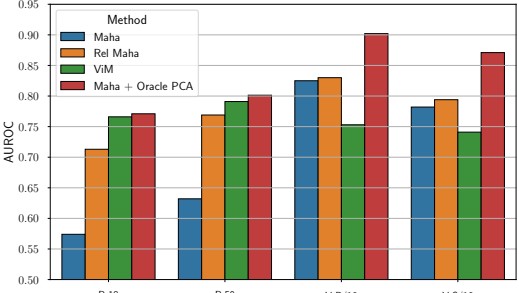

Figure A.2: **Relative Mahalanobis and ViM do not fully address the issue of irrelevant features on ImageNet vs ImageNet-OOD, especially with the more performant ViT models.** In all cases, Mahalanobis with an Oracle PCA performs the best. Except for ResNets, Relative Mahalanobis and ViM offer negligible or negative improvement relative to Mahalanobis. The gap between Maha + Oracle PCA and the best-performing feature-based method is especially large for ViTs.

## A.2    LOGIT-BASED METHODS

Logit-based methods fail when the uncertainty of ID data looks similar to the uncertainty of OOD data. We see an example in Figure A.3, where we find that the model very confidently classifies OOD dogs as ID trucks. In Table A.1, we find that well over half of OOD examples are misclassified as ID even with powerful pre-trained models. Figure A.4 visualizes a failure mode for a ResNet-50 trained on ImageNet, where 'Stripes' is often miscategorized as ID. In Figure A.5, we find that this failure mode is prevalent across a diverse set of models and logit-based OOD-detection methods.

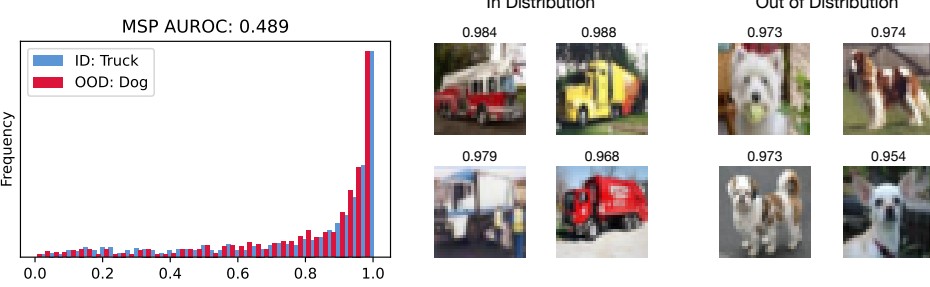

(a) Confidence of ID vs OOD inputs                    (b) Example inputs and model confidence

Figure A.3: **The predictive uncertainty of OOD points may be indistinguishable from ID points.** We train a LeNet5 to classify CIFAR10 automobiles and trucks, and we test the OOD dog class. We see that the model confidence for OOD dogs entirely overlaps with the ID truck class. In this setting, because the uncertainties are identical, no uncertainty-based method would be able to successfully differentiate ID from OOD.

| OOD Dataset | Model | MSP | Max Logit | Entropy | Energy Score |
|---|---|---|---|---|---|
| IN-OOD | ResNet-50 | 0.774 | 0.804 | 0.820 | 0.778 |
| IN-OOD | ResNet-50 DINO | 0.804 | 0.830 | 0.847 | 0.823 |
| IN-OOD | ResNet-34 | 0.807 | 0.824 | 0.838 | 0.809 |
| IN-OOD | ResNet-18 | 0.832 | 0.846 | 0.855 | 0.845 |
| IN-OOD | ViT-S/16 | 0.797 | 0.803 | 0.818 | 0.798 |
| IN-OOD | ViT-S/16 DINO | 0.761 | 0.790 | 0.811 | 0.768 |
| IN-OOD | ViT-B/16 | 0.740 | 0.733 | 0.771 | 0.726 |
| IN-OOD | ViT-B/16 DINO | 0.741 | 0.764 | 0.784 | 0.749 |
| IN-OOD | ViT-B/16 CLIP | 0.764 | 0.776 | 0.805 | 0.726 |
| IN-OOD | ViT-B/14 DINOv2 | 0.658 | 0.621 | 0.638 | 0.610 |
| IN-OOD | ViT-G/14 DINOv2 | 0.562 | 0.448 | 0.450 | 0.469 |
| IN-OOD | ViT-L/14 CLIP | 0.686 | 0.685 | 0.723 | 0.631 |
| IN-OOD | ConvNeXt V2-B | 0.701 | 0.708 | 0.773 | 0.673 |
| IN-OOD | ConvNeXt V2-L | 0.696 | 0.710 | 0.787 | 0.663 |
| Textures | ResNet-50 | 0.662 | 0.544 | 0.522 | 0.594 |
| Textures | ResNet-50 DINO | 0.681 | 0.624 | 0.612 | 0.637 |
| Textures | ResNet-34 | 0.690 | 0.562 | 0.533 | 0.620 |
| Textures | ResNet-18 | 0.710 | 0.571 | 0.527 | 0.643 |
| Textures | ViT-S/16 | 0.672 | 0.579 | 0.506 | 0.593 |
| Textures | ViT-S/16 DINO | 0.612 | 0.400 | 0.363 | 0.521 |
| Textures | ViT-B/16 | 0.586 | 0.544 | 0.573 | 0.521 |
| Textures | ViT-B/16 DINO | 0.531 | 0.351 | 0.307 | 0.437 |
| Textures | ViT-B/16 CLIP | 0.657 | 0.530 | 0.538 | 0.564 |
| Textures | ViT-B/14 DINOv2 | 0.535 | 0.409 | 0.401 | 0.451 |
| Textures | ViT-G/14 DINOv2 | 0.480 | 0.344 | 0.332 | 0.389 |
| Textures | ViT-L/14 CLIP | 0.543 | 0.441 | 0.446 | 0.462 |
| Textures | ConvNeXt V2-B | 0.530 | 0.480 | 0.490 | 0.441 |
| Textures | ConvNeXt V2-L | 0.551 | 0.468 | 0.480 | 0.440 |
| iNaturalist | ResNet-50 | 0.703 | 0.700 | 0.716 | 0.684 |
| iNaturalist | ResNet-50 DINO | 0.644 | 0.594 | 0.598 | 0.619 |
| iNaturalist | ResNet-34 | 0.745 | 0.721 | 0.726 | 0.728 |
| iNaturalist | ResNet-18 | 0.739 | 0.727 | 0.734 | 0.727 |
| iNaturalist | ViT-S/16 | 0.726 | 0.683 | 0.668 | 0.692 |
| iNaturalist | ViT-S/16 DINO | 0.726 | 0.660 | 0.658 | 0.699 |
| iNaturalist | ViT-B/16 | 0.692 | 0.711 | 0.791 | 0.674 |
| iNaturalist | ViT-B/16 DINO | 0.682 | 0.617 | 0.613 | 0.648 |
| iNaturalist | ViT-B/16 CLIP | 0.698 | 0.655 | 0.683 | 0.634 |
| iNaturalist | ViT-B/14 DINOv2 | 0.519 | 0.429 | 0.426 | 0.455 |
| iNaturalist | ViT-G/14 DINOv2 | 0.448 | 0.355 | 0.351 | 0.384 |
| iNaturalist | ViT-L/14 CLIP | 0.593 | 0.547 | 0.566 | 0.522 |
| iNaturalist | ConvNeXt V2-B | 0.634 | 0.638 | 0.712 | 0.589 |
| iNaturalist | ConvNeXt V2-L | 0.627 | 0.624 | 0.704 | 0.563 |

Table A.1: **FPR@95 for OOD detection remains high with popular models.** We record the FPR@95 for the MSP method for 14 models including ResNets, ViTs, and ConvNext V2 models on ImageNet-1K as ID, and Textures, iNaturalist, and ImageNet-OOD as OOD. FPR@95 records how many OOD examples are classified as ID (low uncertainty, false positive) at a threshold where 95% of ID examples are correctly classified (true positive). The average FPR@95 over all models and OOD datasets is 66.5%, thus well over half of OOD examples are classified as ID due to having low uncertainty, and other methods such as max logit, energy score, and entropy all have similar FPR@95s of over 60%.

## A.3 HYBRID METHODS

We find hybrid methods like ViM and Hybrid-Add work well on far-OOD datasets like Textures, where we see noticeable improvement across many models in Figure A.6.

## A.4 EFFECT OF PRE-TRAINING

Miller et al. (2021) showed that there is a strong linear relationship between ID accuracy and OOD generalization on OOD data with covariate shifts, suggesting it is sufficient to focus on improving

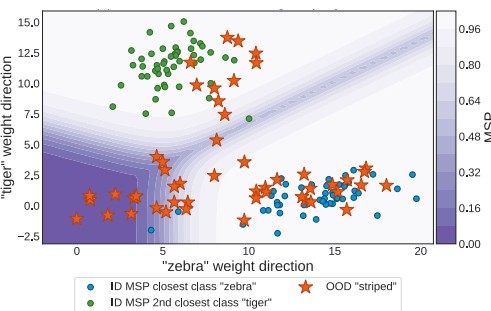

Figure A.4: For a ResNet-50 trained on ImageNet-1k, we see that the model has very high confidence for the OOD class 'Striped', highlighting the difference between label uncertainty and OOD uncertainty.

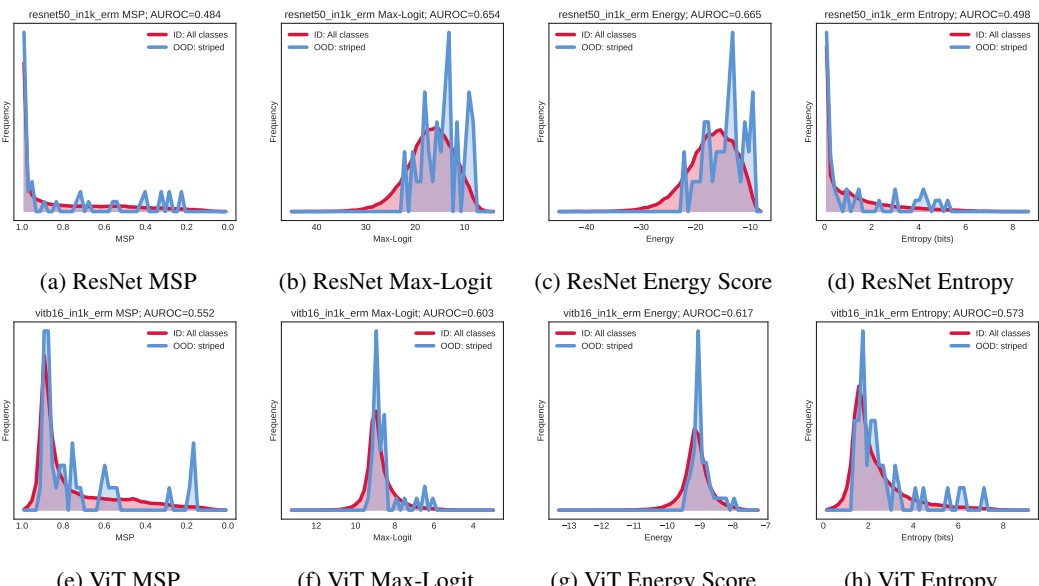

Figure A.5: We plot the distribution of OOD scores on ImageNet-1K ID and Describable Textures 'striped' class OOD data obtained from different OOD detection methods. We discover a systematic failure mode of all methods that utilize logits stemming from the model being overconfident about its predictions on the OOD data. Even though different OOD detection methods have different AUROC numbers, the score distribution plots reveal it is difficult to cleanly separate ID and OOD scores by picking a threshold. We use a ResNet-50 pretrained on ImageNet-1K and use a ViT-B/16 pretrained on ImageNet-1K.

ID accuracy for better robustness. Similarly, we explore the connection between the test accuracy and OOD detection performance. We use ImageNet-1K (IN-1K) as ID data and ImageNet-OOD (IN-OOD) (Yang et al., 2024) and Textures (Cimpoi et al., 2014) as OOD data. We evaluate 54 models covering a wide range of architectures and pretraining methods. In Figure A.7 we plot AUROC of MSP against ImageNet test accuracy. Generally, ID accuracy and AUROC have close to a linear relationship for models with low- to medium-range performance on ImageNet. However, on ImageNet-OOD for models performing around or better than 75% ID accuracy, we observe higher variability in AUROC: for larger-scale highly performant models pre-training data impacts the OOD detection more significantly.

When models are exposed to a diverse set of data during pre-training, they are likely to learn a wide range of features, making it possible for them to differentiate between ID and semantically new classes. There is an edge case when the OOD data is included in pre-training dataset: in Figure A.7 for the best performing ViT and ConvNext models pre-training includes ImageNet-21K, after which they are fine-tuned on ImageNet-1K. Since ImageNet-OOD consists of images from ImageNet-21K

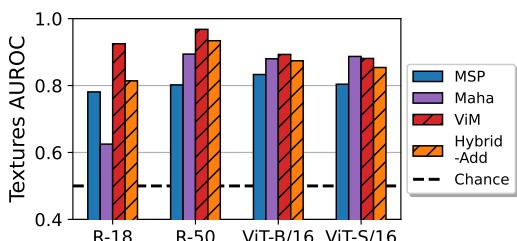

Figure A.6: Hybrid OOD methods outperform logit and feature-based on Textures

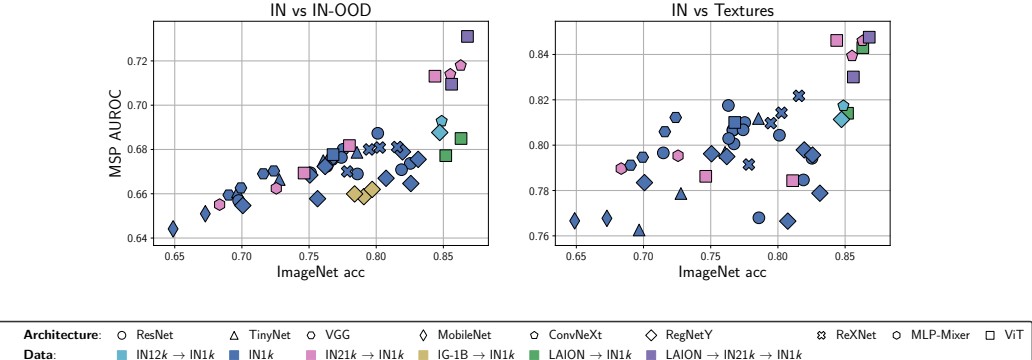

Figure A.7: The impact of the model architecture, pre-training data and objective on OOD detection performance. AUROC of MSP on ImageNet vs ImageNet-OOD (left) and ImageNet vs Textures (right) against ImageNet test accuracy. We observe that improving ImageNet accuracy generally leads to better OOD detection.

which do not semantically overlap with ImageNet-1K classes, we observe a rapid jump in AUROC for these models with negligible variability in ID accuracy. Pre-training on diverse data which includes similar examples to OOD points softens the misspecification of the MSP approach and leads to strong performance.

## A.5 GENERATIVE MODELS

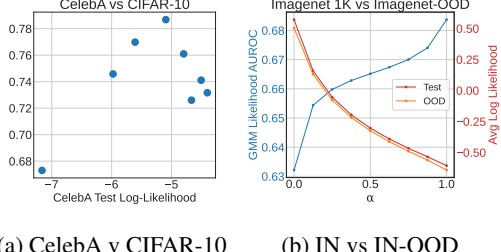

(a) CelebA v CIFAR-10          (b) IN vs IN-OOD

Figure A.8: **Better generative model of ID data can lead to worse OOD detection**. **Left:** RealNVP models achieving better likelihoods on ID CelebA images do not consistently achieve better AUROC for detecting CIFAR-10. **Right:** The Gaussian Mixture Model (GMM) on ResNet-50 features achieves best likelihoods with the empirical covariance of ImageNet features, but achieves best AUROCs for detecting ImageNet-OOD with the identity covariance ($\alpha = 1$). $\alpha$ represents the linear interpolation coefficient towards identity covariance.

**Conceptual limitation of generative models for OOD detection.** Estimating $p(x)$ is different from estimating whether $x$ is more likely to be drawn from some different distribution. Conceptually, for the latter, we would like to compute $p(\text{OOD}|x)$, which by Bayes' rule, is $p(x|\text{OOD})/p(x)$ up to an $x$-independent constant. In general, knowing $p(x)$ tells us nothing about the value of this ratio. $p(x|\text{OOD})/p(x)$ is also invariant to any coordinate transformation on $x$, whereas $p(x)$ is not.

Measuring typicality rather than density is an alternative method for OOD detection. Rather than asking whether a point has a high density, typicality asks whether a point belongs to a region with high probability mass. However, typicality has very similar pathologies compared to density. Consider a common motivating example for typicality: points drawn from a high dimensional Gaussian $\mathcal{N}(0, I)$ in $\mathbb{R}^d$ will have norms $\|x\| = \sqrt{\sum_{i=1}^d x_i^2}$ concentrating around $\sqrt{d}$ by the Law of Large Numbers (LLN). A point at the origin will be considered highly OOD based on typicality, since it has zero norm, yet it has the highest density and will thus be considered highly ID based on the density. But there is no reason why we should judge typicality based on norm rather than other quantities. Consider the average value of $x$ over the dimensions, $\frac{1}{d}\sum_{i=1}^d x_i$, this quantity concentrates around 0 by LLN. Based on this quantity, a point at the origin looks perfectly typical, while a point on a sphere of radius $\sqrt{d}$ looks highly atypical. Therefore, exactly similar to the density, notions of typicality will tend to depend on a subjective choice of how to coarse-grain the input space based on quantities that are most relevant for distinguishing between ID and OOD data. Finally, measures of typicality can also depend on an arbitrary choice of coordinates. For example, $(\epsilon, N)$-typical set (Nalisnick et al., 2020) relies on the differential entropy, which is not invariant to coordinate transformations.

**OOD Detection Requires Coarse-Grained Representations.** In general, every test input we encounter will differ from the ID inputs we have previously seen. However, not all test inputs are considered OOD because we are only concerned with differences in certain essential aspects. When learning a generative model $p(x)$ of the ID data, our goal is not necessarily to capture the distribution of $x$ in its finest details. Instead, for the purpose of OOD detection, it is more appropriate to model the distribution over a coarse-grained representation $h(x)$, which captures the attributes necessary for distinguishing OOD from ID data and ideally nothing more.

Consider an ID dataset consisting of 1000 breeds of dogs and 10 breeds of cats. If our generative model captures the frequency of each individual breed, any dog input we observe will typically be considered 100 times more OOD-like than any cat input based on the likelihood of the generative model. However, if our goal is to detect other animal species and non-animal objects, the likelihood of this model is clearly not aligned with the objective of OOD detection. In this case, it would be more suitable to model only the frequency over the dog and cat categories, which serves as an appropriately coarse-grained representation of the individual breeds. Since the definition of OOD is ultimately user-defined, the correct coarse-grained representation depends on both the dataset and the intended definition of OOD, and it might be challenging to accurately specify even when a definition is known.

In Figure A.8a, we show the test log-likelihoods (normalized by dimension) of RealNVP (Dinh et al., 2016) normalizing flow models of various sizes trained on CelebA images and their AUROCs for detecting CIFAR-10. While models with the lowest test likelihoods on ID data perform poorly for OOD detection, their OOD detection performance does not improve monotonically with their test likelihoods. In fact, the AUROC eventually decreases with improvements in likelihood.

In Figure A.8b, we demonstrate the same phenomenon for a feature-space generative model. We construct a Gaussian Mixture Model (GMM) model of the features produced by a ResNet-50 pre-trained on ImageNet-1K, the ID dataset. To optimize for the likelihood on ID data, we choose the cluster means to be the class-conditional means and use the empirical covariance of all features centered by their respective class means as the covariance of the clusters. This GMM model is precisely the generative model used by the Mahalanobis method (Lee et al., 2018). As we interpolate between the empirical covariance and a trivial identity covariance, the ID test likelihood of this GMM model decreases, yet the AUROC for detecting ImageNet-OOD improves monotonically.

**The Impact of Inductive Biases.** How a generative model assigns density to data unseen during training is highly dependent on their inductive biases. Despite being highly flexible density models, normalizing flows are known to be poor OOD detectors when trained as a generative model over raw images because their inductive biases encourage the model to focus on low-level pixel correlations rather than high-level semantic properties (Kirichenko et al., 2020; Nalisnick et al., 2018). Here, we demonstrate that the same failure mode applies to diffusion models, a distinct class of generative models achieving state-of-the-art image generation (Betker et al., 2023; Saharia et al., 2022).

We use the Diffusion Transformer (DiT), a 256x256-resolution latent diffusion model trained on ImageNet-1K (Peebles and Xie, 2023). We score images based on the diffusion loss, a variance-

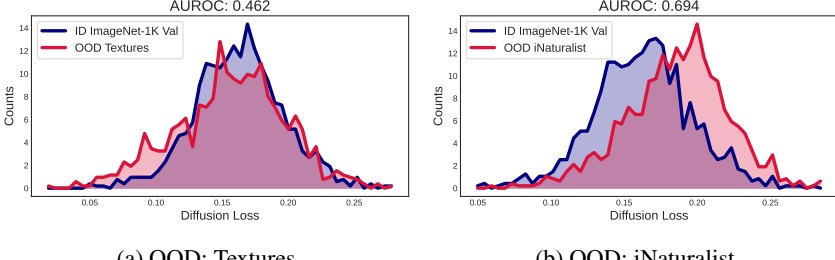

(a) OOD: Textures          (b) OOD: iNaturalist

Figure A.9: **Diffusion Models can fail catastrophically at OOD.** (**a**): Using the diffusion loss, the Diffusion Transformer (DiT) (Peebles and Xie, 2023) fails catastrophically at detecting OOD inputs from the Describable Textures dataset. (**b**): the DiT model does decently at detecting OOD inputs from iNaturalist.

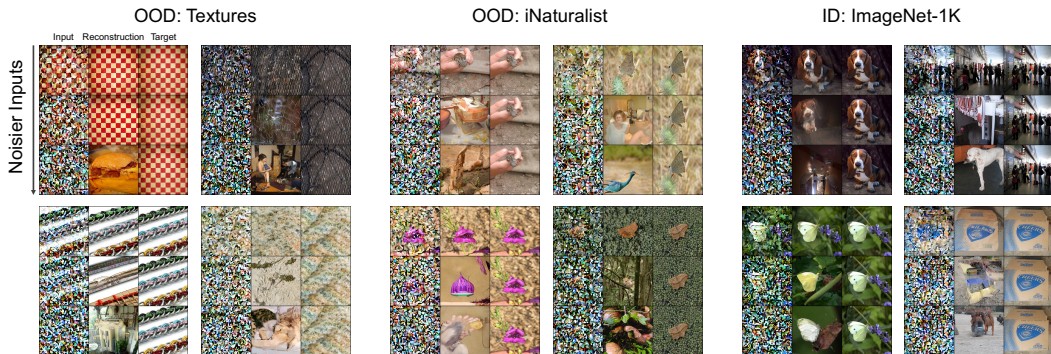

Figure A.10: **Visualization of DiT Reconstruction Error.** A DiT trained on ImageNet-1K often accurately reconstructs noised images from Describable Textures despite never having trained on them. **Left**: Reconstructions of noised Describable Textures images compared to **middle**: iNaturalist images and **right**: ImageNet-1K images.

reduced approximation of the variational lower bound (Kingma et al., 2021; Ho et al., 2020). In Figure A.9, we show the DiT fails catastrophically in detecting OOD data from Describable Textures but achieves decent performance in detecting OOD data from iNaturalist.

In Figure A.10, we qualitatively show that a 256x256 DiT trained on ImageNet-1K often accurately reconstructs noised images from Describable Textures despite never having trained on them. We add noise to the inputs corresponding to the diffusion timesteps at 49, 98, 147 out of 249, where higher timesteps are more noisy.

## A.6 OUTLIER EXPOSURE

Training a model with outlier exposure is effective for improving OOD detection, and we see that performance is improved for most OOD problems with semantic shifts in Figure A.11

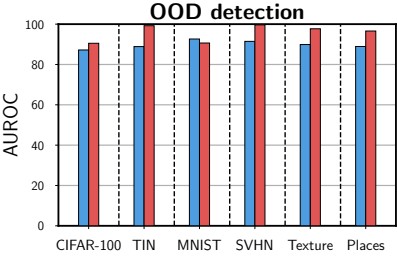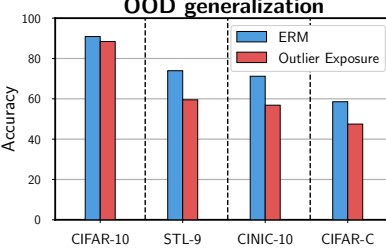

Figure A.11: Training a ResNet-18 with outlier exposure improves OOD detection for semantic shift datasets but hurts OOD generalization over covariate shifts.

# B IMPLEMENTATION DETAILS

## B.1 OUTLIER EXPOSURE EXPERIMENT.

On Figure 6, we compare OE model to the baseline ERM training in OOD detection (left panel) and OOD generalization (right panel). For semantic shift detection, we use CIFAR-100, Tiny ImageNet, MNIST, SVHN, Textures (Cimpoi et al., 2014), and Places365 (Zhou et al., 2014). For OOD generalization we evaluate on STL-10 (Coates et al., 2011), CINIC-10 (Darlow et al., 2018) and CIFAR-10-C (Hendrycks and Dietterich, 2019).

We adapt OpenOOD codebase (Zhang et al., 2023; Yang et al., 2022) to train ResNet-18 with baseline ERM training and Outlier Exposure (Hendrycks et al., 2018) and evaluate models on OOD detection. We train models for 100 epochs with batch size 128 for ID data and batch size 256 for the outlier dataset, SGD with momentum and initial learning rate 0.1 and weight decay $5 \times 10^{-4}$, and we set the coefficient before the OE loss to $alpha = 0.5$ (overall, we use standard training hyper-parameters as in Zhang et al. (2023)). For Figure 6, we run both methods with 3 random seeds and report the average performance. To evaluate the model on STL-10, we only use the 9 classes which overlap with CIFAR-10 classes and drop the class "monkey" not present in CIFAR-10 (thus, the evaluation is marked as STL-9 in Figure 6). For CIFAR-C, we report the average accuracy across 15 corruptions (Gaussian Noise, Shot Noise, Impulse Noise, Defocus Blur, Glass Blur, Motion Blur, Zoom Blur, Snow, Frost, Fog, Brightness, Contrast, Elastic transform, Pixelate, JPEG Compression).

## B.2 EVALUATING PRE-TRAINED MODELS.

We evaluate 54 models from the `timm` and `torchvision` libraries, including 9 different arhcitecture types: ResNet, TinyNet, VGG, MobileNet, ConvNeXt, RegNetY, ReXNet, MLP-Mixer, and ViT; and 6 different pre-training data setups: training on IN-1K from scratch, pre-training on IN-21K and fine-tuning on ON-1K, pre-training on IN-12K (a subset of IN-21K) and fine-tuning on IN-1K, CLIP (Radford et al., 2021) pre-training on LAION and fine-tuning on IN-1K, CLIP pre-trainig on LAION and further fine-tuning on IN-21K and then IN-1K, and Instagram-1B pre-training and further IN-1K fine-tuning of SEER models (Goyal et al., 2021).

## B.3 SCALING EXPERIMENTS

We benchmark the following models to demonstrate impact of scale in Figure 4:

1. ResNet-18 trained on ImageNet-1k

2. ResNet-34 trained on ImageNet-1k

3. ResNet-50 trained on ImageNet-1k

4. ViT-S/16 trained on ImageNet-1k

5. ViT-B/16 trained on ImageNet-1k

6. ViT-S/16 trained on ImageNet-1k with DINO

7. ViT-B/16 trained on ImageNet-1k with DINO

8. ViT-B/16 trained with CLIP

9. ViT-L/14 trained with CLIP

10. ViT-B/16 pretrained on CLIP, finetuned on ImageNet-1k

11. ViT-B/14 trained on 142M images with DINOv2

12. ViT-G/14 trained on 142M images with DINOv2

