# OpenReview forum: "Pathologies of Out-of-Distribution Detection"
_ICLR.cc/2025/Conference — Submitted to ICLR 2025_

### Official Review · Reviewer_S8wV · 2024-10-27

**Soundness:** 2
**Presentation:** 2
**Contribution:** 2
**Rating:** 5
**Confidence:** 3

**Summary:**

This paper critically re-examines the popular family of OOD detection procedures, which rely on supervised learning to train models with in-distribution data and then use the models’ predictive uncertainty or features to identify OOD points. The analysis reveals deep-seated pathologies. It argues that these procedures are fundamentally answering the wrong question for OOD detection, with no easy fix. Uncertainty-based methods incorrectly conflate high uncertainty with being OOD, and feature-based methods incorrectly conflate far feature-space distance with being OOD. Moreover, there is no reason to expect a classifier trained only on in-distribution classes to be able to identify OOD points. It shows how these pathologies manifest as irreducible errors in OOD detection and identifies common settings where these methods are ineffective. Additionally, it shows that interventions to improve OOD detection such as feature-logit hybrid methods, scaling of model and data size, Bayesian uncertainty representation, and outlier exposure also fail to address the fundamental misspecification.

**Strengths:**

1. This paper introduces a fresh perspective by highlighting that many popular OOD detection methods – whether supervised or generative – are not effectively addressing the core question: Is this unlabeled point from a different distribution?

2. This paper provides concrete examples illustrating how widely-used OOD detection techniques fundamentally miss the mark for true OOD detection. The paper further demonstrates that both feature-based and logit-based methods inherently suffer from irreducible errors, limiting their effectiveness in this context.

3. This paper is well-structured, with clear explanations, making it easy to follow.

**Weaknesses:**

1. While the paper argues that popular OOD detection methods do not directly address the question of whether an unlabeled point is from a different distribution, these methods have demonstrated strong performance across established benchmarks. It remains unclear whether adopting the approach suggested in the paper would lead to meaningful improvements in performance, raising questions about the practical utility of the findings.

2. The experiments are relatively simple and may not sufficiently support the claims. For instance, the study relies on a small-scale experiment using a LeNet-5 model trained to classify automobiles and trucks from CIFAR-10 to argue that OOD examples often exhibit low uncertainty. However, it is unclear whether these findings generalize to more complex architectures, such as ResNet, or to larger, real-world datasets.

3. While the paper provides a critical analysis of existing OOD detection methods, it does not propose new approaches to address the identified limitations or offer concrete suggestions for overcoming the challenges it highlights. This limits the practical contribution and leaves open questions about how to improve OOD detection in light of the paper’s critiques.

**Questions:**

1. Given that existing OOD detection methods perform well on benchmarks, how do the authors envision a method that directly addresses the question “Is this unlabeled point from a different distribution?” improving current performance? Could the authors provide further theoretical insights or experimental scenarios to clarify when and why their proposed perspective might offer an advantage?

2. Have the authors considered evaluating their findings with more advanced architectures, such as ResNet, and on larger or more diverse datasets beyond CIFAR-10? Doing so could help demonstrate the generalizability of the claims.

3. While the paper offers a valuable critique of existing methods, have the authors considered proposing potential approaches or heuristics to address the identified limitations? Even if speculative, some guidance or future directions could make the paper’s contributions more actionable for researchers looking to build on the work.

---

> ### Author Response · Authors · 2024-11-14
>
> Thank you for your review. We would like to clarify a few important misunderstandings.
>
> While many OOD detection methods are effective for smaller datasets like CIFAR-10, our paper demonstrates why these fundamental limitations prevent robust detection on more challenging problems like ImageNet-1k. Yang et al. [1] benchmark the performance of many SoTA OOD detection methods such as ViM, Max-Cosine, KNN, ASH-B, ReAct, etc, and find that the best of these methods only achieve an AUROC of 0.807 on ImageNet-1k vs ImageNet-OOD, with some methods scoring an AUROC of 0.593. These methods perform even more poorly under covariate shift, often achieving worse than random chance. To demonstrate that OOD detection cannot easily be solved by scaling up model and data size, we show in Figure 4 that even a ViT-G/14 DINOv2 pre-trained on internet-scale data, indicated by the rightmost points, still has significant irreducible error. Therefore, OOD detection methods still have considerable room for improvement, and optimal performance can only be achieved by rethinking the approach to OOD detection and moving away from using supervised models which are mis-specified for the problem.
>
> [1] Yang et al, ImageNet-OOD: Deciphering Modern Out-of-Distribution Detection Algorithms
>
> We would also like to clarify that our claims in the paper do not rely on empirical examples; instead, we provide a conceptual understanding of the fundamental limitations of how these methods operate, and we use illustrative examples to show how these conceptual pathologies lead to suboptimal performance in practice. Furthermore, we actually do provide extensive benchmarks to support each of our claims:
> - To demonstrate the pathologies of feature-based methods (section 4.1), we provide experiment results using four different models: ResNet-18, ResNet-50, ViT-S/16, and ViT-B/16. We use ImageNet-1k as our ID data, and we test on three different OOD datasets: ImageNet-OOD, Textures, and iNaturalist. We also use a ResNet-18 trained on CIFAR-10 for visualization purposes, since smaller feature dimensions and fewer classes lead to clearer 2D plots; furthermore, we also visualize failure modes for ResNet-50 on ImageNet-1k.
> - To understand how these feature-based pathologies scale with model and dataset size (Section 5.1 and Figure 4), we benchmark 12 different models of varying sizes and training methods, as listed in Appendix B.3.
> - For pathologies of logit-based methods (Section 4.2 and Figure 3), we show results for ResNet-18, ResNet-50, ViT-S/16, ViT-B/16, trained on ImageNet, and ViT-G/14 trained with DINOv2. We also use a LeNet-5 trained on CIFAR-10 for visualization purposes.
> - To understand the scaling behaviors of logit-based methods (Section A.4 and Figure A.7), we evaluate 54 models from timm and torchvision, with 9 different architectures and 6 different training/pre-training setups, listed in Appendix B.2.
>
> While these experimental setups were included in the original paper, we have revised the text to highlight the diversity of the models and datasets we benchmarked. We hope this addresses your concerns about generalizability.
>
> We do not propose yet another method to benchmark; instead, our work explicitly identifies the problem misspecification of the existing approaches which utilize supervised models trained on ID data. We do find that this misspecification is slightly alleviated when the model is first pretrained on very diverse datasets, as the model is then incentivized to learn robust distinguishing features. As such, we encourage practitioners to think more deeply about the data that the model has been exposed to, and we recommend using datasets that are as large and diverse as possible for general OOD detection. However, the goals remain fundamentally unaligned and there are still irreducible sources of error from irrelevant features as shown in Figure 4, so we conclude that more work on OOD detection is needed to develop procedures which directly address the problem.
>
> Prescriptive measures are not necessary for a paper to have scientific merit, and we believe that deepening understanding, critically examining limitations, and prompting further research are also important contributions. Many papers of this variety have been very successful at ICLR, ICML, and NeurIPS. To name a few:\
> “The marginal value of adaptive gradient methods in machine learning” (NeurIPS 2017, 1278 citations)\
> “Understanding deep learning requires rethinking generalization” (ICLR 2016 best paper)\
> “How Good is the Bayes Posterior in Deep Neural Networks Really?” (ICML 2020, 373 citations)\
> “Bayesian Model Selection, the Marginal Likelihood, and Generalization” (ICML 2022 best paper)\
> “If Influence Functions are the Answer, Then What is the Question?” (NeurIPS 2022)
>
> We would appreciate it if you would consider re-evaluating our paper in light of these clarifications, and we would be happy to answer any further questions.

---

> ### Author Response · Authors · 2024-11-20
> **Author Response to S8wV**
>
> We hope our response has addressed your questions, and we would really appreciate further engagement with our work. Given the nature of the misunderstandings, we hope you can re-evaluate our paper in light of our corrections.
>
> In response to your feedback, in order to further demonstrate instances where OOD examples have low uncertainty, we now include **additional experiments for the rebuttal**, shown in lines 294-300 and Appendix A.2. Specifically, we benchmark 14 different models including ResNets, ViTs, and ConvNeXt V2 on the realistic setting where ImageNet-1K is ID, and we evaluate the performance of MSP for three different OOD datasets: Textures, iNaturalist, and ImageNet-OOD. We record the FPR@95, which indicates how many OOD examples are incorrectly classified as ID due to their low uncertainty (false positive), at a threshold where 95% of ID examples are correctly classified (true positive). The average FPR@95 over all models on all three OOD datasets is 66.5%; thus well over half of OOD examples are classified as ID due to their low uncertainty, and other methods such as max logit, energy score, and entropy all have similar FPR@95s over 60%. These experiment results demonstrate that OOD inputs having low uncertainty is a common pathology across many models and datasets.
>
> We would also like to emphasize that the primary contribution of our paper is to deepen the community's understanding of the limitations and pitfalls of current methodologies. While we agree that it is necessary to develop new approaches for OOD detection, resolving these limitations ourselves would require a significantly broader scope than we intend for our work. In fact, comprehensively addressing the identified pathologies would likely require foundational progress to the field of OOD detection itself. Such a contribution would extend well beyond the typical scope of work in OOD, where many papers focus on incremental improvements to existing methods within established families of approaches. Furthermore, it is worth nothing that the current state of OOD detection reflects years of iterative research by the community, and it may be overly ambitious to expect one section of our paper to offer a complete resolution. Instead, we hope our work will inspire future efforts towards more principled approaches.
>
> Please let us know if you have any further questions or comments, and we look forward to your response.

---

> > ### Comment · Reviewer_S8wV · 2024-11-25
> >
> > Thank you for your detailed responses. Some of my concerns have been addressed. I will raise my scores. However, I still think it is important to propose new approaches to address the identified limitations or offer concrete suggestions for overcoming the identified challenges.

---

### Official Review · Reviewer_7Rbf · 2024-11-03

**Soundness:** 3
**Presentation:** 3
**Contribution:** 3
**Rating:** 6
**Confidence:** 4

**Summary:**

Authors claim that current OOD detection methods, often based on supervised models and predictive uncertainty, are fundamentally misguided, as they attempt to answer the wrong questions about whether a sample truly belongs to the training distribution.

Currently, common OOD detection methods involve training a supervised model on in-distribution data and subsequently using the model’s predictive uncertainty or features to detect OOD instances. These approaches are typically divided into two main streams
(1) Feature-based methods and (2) Logit-based methods.

However, authors argue that these methods inherently answer the wrong question because they do not measure distribution membership but rather whether a sample leads to unexpected model representations. First, feature-based methods fail when features of OOD and in-distribution data overlap or are indistinguishable, leading to irreducible errors in OOD detection. Second, logit-based methods with high label uncertainty does not reliably indicate OOD samples, as in-distribution samples with high label ambiguity may appear as OOD. This conflation leads to missed detections or incorrect OOD classifications.

Also, alternative attempts to improve OOD detection through scaling, hybrid models, or outlier exposure fall short, as these interventions still rely on the flawed assumptions underlying feature- and logit-based detection.
(1) Scaling model size: Increasing model and dataset size does not fundamentally address the limitations in feature separation for OOD detection.
(2) Hybrid methods: Combining features and logits may show slight improvements but does not resolve the underlying misalignment with true OOD detection which adressed above.
(3) Outlier Exposure: Training with outlier samples to simulate OOD data may improve detection but reduces generalization performance on covariate shifts, highlighting a trade-off between detection and generalization.
(4) Methods based on epistemic (Bayesian) uncertainty, which should theoretically improve with increased data, fail as they conflate uncertainty about the model’s knowledge with uncertainty over whether a sample is OOD.
(5) Generative models, which compute likelihoods of training data, often fail in OOD detection due to their inability to differentiate low-likelihood in-distribution data from high-likelihood OOD data.

**Strengths:**

This paper tried to adress its critical examination of the conceptual limitations of widely used OOD detection methods. By challenging the assumptions underlying feature- and logit-based OOD techniques, the authors reveal fundamental pathologies—such as the conflation of high uncertainty with out-of-distribution status. This critique goes beyond incremental improvements and instead questions the foundational approach, which could reshape the field’s direction. Overall the paper is well written in clear presentations with a topic should be high interest in the field.

**Weaknesses:**

Although, I agree that there is a fundamental pathologies in OOD detection which mentioned in the strength of the paper, I still have some concerns regarding the paper.

1. OOD features that are indistinguishable from ID features may due to the small model representation space. Larger models with various classes may learn detailed representation that can distinguish.

2. Recent OOD methods have devided definition of OOD score and uncertainty. It might seem simmilar, but OOD scores are for detecting whether the input is OOD or ID, not serving as the label is certain or not. This refers there can be a sample with low label uncertainty with high OOD scores (and this is the reason that most SOTA ood detection methods outperform MSP in tables).

**Questions:**

Interesting paper with nice contributions. You may want to address the weaknesses I listed.

---

> ### Author Response · Authors · 2024-11-14
> **Author Response to 7Rbf**
>
> Thank you for your review! We are glad that you recognize our work “goes beyond incremental improvements," and we address your concerns below.
>
> We explore the impacts of scaling model size and pre-training data in Figure 4 and Figure A.7. We see in Figure 4 that even for the rightmost point representing ViT-G/14 DINOv2 pre-trained on internet-scale data, there is still significant irreducible error (the difference between the perfect performance of 1.0 and the purple triangle) for feature-based methods. This irreducible error can be further broken down into “indistinguishable features” (the difference between perfect 1.0 and the green star), as well as “irrelevant features” (the gap between the blue triangle and the purple triangle). These results demonstrate that the crux of the pathology remains unchanged even for SoTA models and pre-training procedures, although we do see that the larger models do have fewer instances of indistinguishable features compared to smaller ones.
>
> While there have been many OOD detection methods which do not purely rely on uncertainty through MSP, we show that logit-based methods also suffer from the pathologies we described. In Figure A.5, we demonstrate that max-logit, energy score, and entropy all fall victim to the same failure mode where the OOD scores for ID and OOD data entirely overlap. Because a supervised model trained only on ID data is not incentivized towards any behavior for OOD data, there is no guarantee that the logits can be used to distinguish ID and OOD.
>
> Furthermore, in Section 5.2, we also explore SoTA methods, like ViM, which combine feature and logit-based methods to compute the OOD score. We find that these hybrid methods also fail to address the fundamental pathologies caused by the model misspecification. For instance, in the cases where ID and OOD features are indistinguishable, hybrid methods would not be able to provide any benefits.
>
> We hope this has addressed your questions, and we are happy to engage with any further comments.

---

> > ### Comment · Reviewer_7Rbf · 2024-11-25
> >
> > I want to thank the authors for their detailed responses. All the questions I've addressed have been solved. Though I will maintain my current score.

---

### Official Review · Reviewer_NHyc · 2024-11-04

**Soundness:** 3
**Presentation:** 4
**Contribution:** 1
**Rating:** 5
**Confidence:** 3

**Summary:**

The paper argues that OOD detection is fundamentally misspecified. It provides a review of
existing OOD techniques and conducts mini-experiments to demonstrate that each suffers
from fundamental flaws arising from the way OOD detection is framed.

**Strengths:**

A substantive assessment of the strengths of the paper, touching on each of the following
dimensions: originality, quality, clarity, and significance. We
encourage reviewers to be broad in their definitions of originality and significance. For
example, originality may arise from a new definition or problem
formulation, creative combinations of existing ideas, application to a new domain, or
removing limitations from prior results. You can incorporate Markdown
and Latex into your review. See https://openreview.net/faq (https://openreview.net/faq).

The paper is well presented, easy to follow, and makes a well argued case. The authors
conduct mini-experiments to demonstrate pathologies across a wide range of OOD methods
and intend to release code publicly to reproduce examples. The topic is relevant to the
community and raises awareness of the need to clarify the framing and purpose of OOD
detection.

**Weaknesses:**

A substantive assessment of the weaknesses of the paper. Focus on constructive and
actionable insights on how the work could improve towards its stated
goals. Be specific, avoid generic remarks. For example, if you believe the contribution lacks
novelty, provide references and an explanation as evidence; if you
believe experiments are insu&quot;cient, explain why and exactly what is missing, etc.

While the paper is well presented and provides a compelling argument, it&#39;s not clear what
the new research contribution is, and as such I cannot recommend it for acceptance in the
main research track at ICLR. I&#39;d encourage submitting as a review article to a journal or as a
position paper to a workshop, but without a significant research contribution I can&#39;t
recommend it for the main research track.
Additional related work:
* Fahim Tajwar, Ananya Kumar, Sang Michael Xie, Percy Liang, &quot;No True State-of-the-Art?
OOD Detection Methods are Inconsistent across Datasets&quot; 2021
https://arxiv.org/abs/2109.05554
* Damien Teney, Yong Lin, Seong Joon Oh, Ehsan Abbasnejad &quot;ID and OOD Performance
Are Sometimes Inversely Correlated on Real-world Datasets&quot; NeurIPS 2023
https://arxiv.org/pdf/2209.00613
Minor:
* Figure 1 doesn&#39;t seem to be referenced in the main text
* Line 89: If taking arg max of a function with respect to K, then I&#39;d expect K to appear
somewhere in the function. I believe the arg max should be with respect to index i ∈ {1, .., K}
and the index shown in the function as either y_i or y = i.
* line 99: features or features -&gt; features of features?
* Line 344: OO -&gt; OOD

**Questions:**

Are the authors claiming a new finding or novel contribution in relation to any specific
experiment, or are these primarily intended to demonstrate known limitations as evidence to
support the paper&#39;s main argument?

---

> ### Author Response · Authors · 2024-11-14
> **Author Response to NHyc**
>
> Thank you for your review and references. We have updated the text to fix the minor issues you pointed out, and we address your concerns below.
>
> Our paper focuses on the problem of OOD detection of semantic shift, where new and unseen classes appear at test-time. We specifically address methods which use the features or logits of supervised models to differentiate between ID and OOD data. Therefore, while problems like OOD generalization for covariate shifts addressed by Teney et al [1] are closely related, they are outside the scope of this paper.
>
> [1] Teney et al, ID and OOD Performance Are Sometimes Inversely Correlated on Real-world Datasets
>
> We demonstrate significant research contributions by being the first to explore why entire families of OOD detection approaches, such as logit-based and feature-based methods, are fundamentally flawed. Because there are many works which criticize specific OOD detection procedures, it may seem that our work lacks novelty. However, many SoTA OOD detection methods focus on adaptations within these existing families, often implying the opposing stance that improved methodology can address these innate limitations.
>
> For instance, in the prior work you referenced by Tajwar et al [2], the authors do address limitations of feature and logit-based methods. However, they focus on understanding the tradeoffs between the methods, and they propose a new feature-based method for low-data regimes, suggesting that choosing the best feature or logit-based method is sufficient.  In contrast, we discover that even these novel methods, or any other logit-based or feature-based methods, will have fundamental limitations that arise from the misspecification of supervised training. Moreover, their results utilize toy datasets and low-resolution datasets like CIFAR-10, while we demonstrate that these pathologies exist even with sophisticated models and large datasets. Thank you for the reference; we have added this discussion to our paper. Other OOD detection papers [3, 4, 5] attribute the failures of OOD detection to a weakness in the specific method and propose a novel method within the same family of approaches. No prior work has explored the inherent pathologies of OOD detection methods which cannot be mitigated by improved methodology or method selection.
>
> [2] Tajwar et al, No True State-of-the-Art? OOD Detection Methods are Inconsistent across Datasets\
> [3] Sun et al, Out-of-Distribution Detection with Deep Nearest Neighbors\
> [4] Wei et al, Mitigating Neural Network Overconfidence with Logit Normalization\
> [5] Wang et al, ViM: Out-Of-Distribution with Virtual-logit Matching\
>
> To this extent, we provide many research contributions which demonstrate the limitations of these methods. We demonstrate the existence of irreducible error for all feature-space methods regardless of methodology, which was previously unexplored and unquantified. Furthermore, we illustrate that this error does not disappear even as we increase model size and data size, disapproving the effectiveness of a commonly proposed fix for OOD detection. Similarly, we also identify sources of irreducible error for logit-based methods, even at scale. We further illuminate the pathologies of potential remedies such as outlier exposure, Bayesian inference, introducing an OOD class, and generative modeling. Although our findings may not seem surprising at first glance (e.g. it seems obvious that feature-based methods will not work for indistinguishable features), our paper provides the important contribution of understanding the extent to which these pathologies occur, even for large models trained on internet-scale data.
>
> We do discuss how pre-training helps reduce the model misspecification issues raised in our paper. We also consider effects of combining various procedures, and the different limitations of generative models. Also, given the generality of our paper, a prescription for a principled general resolution to OOD detection is out of scope and not possible as e.g. a section of a paper. Moreover, prescriptive measures are not necessary for a paper to have strong scientific merit, and we believe that deepening understanding, critically examining limitations, and prompting further research are also important contributions. Many papers of this variety have been very successful at ICLR, ICML, and NeurIPS. To name a few:\
> “The marginal value of adaptive gradient methods in machine learning” (NeurIPS 2017, 1278 citations)\
> “Understanding deep learning requires rethinking generalization” (ICLR 2016 best paper, 7552 citations)\
> “How Good is the Bayes Posterior in Deep Neural Networks Really?” (ICML 2020, 373 citations)\
> “Bayesian Model Selection, the Marginal Likelihood, and Generalization” (ICML 2022 best paper)\
> “If Influence Functions are the Answer, Then What is the Question?” (NeurIPS 2022)
>
> We hope this has addressed your questions, and we are happy to engage with any further comments.

---

> > ### Comment · Reviewer_NHyc · 2024-11-29
> >
> > I thank the authors for their response and confirm that the minor issues raised in my review have been fixed in the latest revision. Nevertheless, I feel this paper might not align with the focus of the main research track of ICLR, which is for presenting and publishing cutting-edge research.
> >
> >
> > I acknowledge that it is not necessary for a paper focussed on critically examining existing methodologies to propose a new solution, and it is refreshing to see the authors take a step back. That said, in these cases, I'd expect to see a novel theoretical framework or high degree of mathematical rigour, as per the examples the authors cite in their response.
> >
> >
> > The paper's main contribution seems to be in synthesising key issues with OOD detection that are hinted at throughout the literature (albeit indirectly) and drawing them together in one consistent work that directly acknowledges the underlying problems and is backed up by a set of experiments that demonstrate the issues on a variety of datasets and models. I believe this is a valuable contribution and would love to see it published, but I don't think it is well suited to the main research track.

---

> ### Author Response · Authors · 2024-11-20
> **Author Response to NHyc**
>
> We hope our response has addressed your questions, and we would really appreciate further engagement with our work. Given the nature of the misunderstandings, we hope you can re-evaluate our paper in light of our corrections.
>
> We would also like to emphasize that the primary contribution of our paper is to deepen the community's understanding of the limitations and pitfalls of current methodologies. While we agree that it is necessary to develop new approaches for OOD detection, resolving these limitations ourselves would require a significantly broader scope than we intend for our work. In fact, comprehensively addressing the identified pathologies would likely require foundational progress to the field of OOD detection itself. Such a contribution would extend well beyond the typical scope of work in OOD, where many papers focus on incremental improvements to existing methods within established families of approaches. Furthermore, it is worth nothing that the current state of OOD detection reflects years of iterative research by the community, and it may be overly ambitious to expect one section of our paper to offer a complete resolution. Instead, we hope our work will inspire future efforts towards more principled approaches.
>
> Please let us know if you have any further questions or comments, and we look forward to your response.

---

### Official Review · Reviewer_K5mb · 2024-11-04

**Soundness:** 2
**Presentation:** 3
**Contribution:** 1
**Rating:** 3
**Confidence:** 4

**Summary:**

The paper presents a critical analysis on prior approaches for out-of-distribution (OOD) detection. The central argument of the paper is that prior approaches cannot detect certain categories of OOD samples, due to limited information regarding OOD samples at test time. The paper demonstrates failure modes for common categories of OOD detection methods such as feature-based, logit-based, and uncertainty-based methods, with concrete empirical evidence and illustrations.

**Strengths:**

1. The paper is overall well written with clear figures and illustrations.
2. The failure modes for different categories of OOD detection methods are clearly demonstrated with examples.

**Weaknesses:**

1. **Limited New Insights and Lack of Technical Depth**: The paper falls short in presenting new insights and lacks significant technical depth. For example:

- Section 4.1: The paper states that “no feature-based method can correctly detect these OOD inputs that have indistinguishable features from ID” (L217) and demonstrates this claim with examples. However, this observation is already well-established in the literature and is almost self-evident given the definition of feature-based OOD detection. Instead of demonstrating the existence of OOD samples that share features with ID (which is an expected finding), a deeper analysis of the root causes—such as the training method, model architecture, or the impact of pre-training—would provide more value.
- Section 4.2: The claim that “OOD examples often have low uncertainty” is supported by an experiment using only LeNet-5 to classify automobiles and trucks from CIFAR-10. The limited scope of this experiment is insufficient to support a generalized conclusion that OOD examples often exhibit low uncertainty.
- Section 5.6: The failure mode of generative models is demonstrated solely with a simplified two-class Gaussian example and lacks accompanying experiments. The relevance of this simple demonstration to recent advancements in generative models remains unclear.

2. **Lack of Scientific Rigor**: The paper lacks scientific rigor in most of its experimental sections. Each section follows a similar format where a failure mode is claimed (often an obvious one) and then demonstrated with a basic experiment (e.g., “To demonstrate feature overlap, we train a ResNet-18 on a subset of CIFAR-10 classes,” L247). These conclusions are drawn from experiments involving only a single model and dataset, leaving open questions about the generalizability of the findings across different training and evaluation settings.


Despite these issues, I acknowledge the educational value the paper provides by illustrating failure cases across different categories of OOD detection methods. I recommend that the authors consider submitting this work to a blog post track, where it could serve as a useful resource for educational purposes.

**Questions:**

See above

---

> ### Author Response · Authors · 2024-11-14
> **Author Response to K5mb**
>
> Thank you for your review. We would like to clarify a few important misunderstandings.
>
> Because there are many works which in some way appear critical of OOD detection procedures, it may seem, superficially, as if our work, which is also critical of OOD detection, lacks novelty. But, diving beneath the surface, this characterization could not be further from the truth. These prior works are focused on adaptations of a family of approaches for better benchmark numbers, not on explaining why a paradigm for OOD detection is fundamentally flawed. For instance, there has been no prior analysis on the inevitable existence of indistinguishable and irrelevant features for feature-based methods, and many papers assume OOD examples will be far from the data without further exploration [1, 2, 3]. Therefore, although some of our findings may not seem surprising at first glance (e.g. it seems obvious that feature-based methods will not work for indistinguishable features), our paper provides the important contribution of understanding the extent to which these pathologies occur, even for large models trained on internet-scale data.
>
> [1] Sun et al, Out-of-Distribution Detection with Deep Nearest Neighbors\
> [2] Ming et al, How to Exploit Hyperspherical Embeddings for Out-of-Distribution Detection?\
> [3] Park et al, Understanding the Feature Norm for Out-of-Distribution Detection
>
> Furthermore, our claims in the paper do not rely on empirical examples; instead, we provide a conceptual understanding of the fundamental limitations of how these methods operate, and we use illustrative examples to show how these conceptual pathologies lead to suboptimal performance in practice. For instance, our section about generative models uses a simple example to explicitly demonstrate how the goals of generative modeling are fundamentally misaligned with the goals of OOD detection; therefore, merely improving the quality of the generative model does nothing to solve the pathology. Furthermore, we actually do provide extensive benchmarks to support each of our claims in the paper:
> - To demonstrate the pathologies of feature-based methods (Section 4.1 and Figure 2), we provide experiment results using four different models: ResNet-18, ResNet-50, ViT-S/16, and ViT-B/16. We use ImageNet-1k as our ID data, and we test on three different OOD datasets: ImageNet-OOD, Textures, and iNaturalist. We also use a ResNet-18 trained on CIFAR-10 for visualization purposes, since smaller feature dimensions and fewer classes lead to clearer plots when we only have 2D images; however, we also visualize failure modes for ResNet-50 and ImageNet-1k.
> - To understand how these feature-based pathologies scale with model and dataset size (Section 5.1 and Figure 4), we benchmark 12 different models of varying sizes and training methods, as listed in Appendix B.3.
> - For pathologies of logit-based methods (Section 4.2 and Figure 3), we show results for ResNet-18, ResNet-50, ViT-S/16, ViT-B/16, trained on ImageNet, and ViT-G/14 trained with DINOv2. We also use a LeNet-5 trained on CIFAR-10 for visualization purposes.
> - To understand the scaling behaviors of logit-based methods (Section A.4 and Figure A.7), we evaluate 54 models from timm and torchvision, with 9 different architectures and 6 different training/pre-training setups, listed in Appendix B.2.
> - For pathologies of generative models (Section 5.6 and Appendix A.5), we use normalizing flows of eight different sizes trained on CelebA. We also use Diffusion Transformers trained on ImageNet-1k to demonstrate the same pathologies.
>
> While these experimental setups were included in the original paper, we have revised the text to highlight the diversity of the models and datasets we benchmarked. We hope this addresses your concerns about scientific rigor.
>
> Although we demonstrate the impacts of model size, architecture, and training method throughout the paper with the experiments listed above, and we explicitly analyze the effect of pre-training in Appendix A.4, one of our key claims is that these pathologies cannot be resolved by simply improving the model or the training procedure. We see in Figure 4 that even for a ViT-G/14 DINOv2 pre-trained on internet-scale data, there is still significant irreducible error (the difference between the perfect performance of 1.0 and the purple triangle) for feature-based methods. This result demonstrates that the crux of the pathology remains unchanged even for SoTA models and pre-training procedures.
>
> We would appreciate it if you would consider re-evaluating our paper in light of these clarifications, and we would be happy to engage with any further questions.

---

> ### Author Response · Authors · 2024-11-20
> **Author Response to K5mb**
>
> We hope our response has addressed your questions, and we would really appreciate further engagement with our work. Given the nature of the misunderstandings, we hope you can re-evaluate our paper in light of our corrections.
>
> In response to your feedback, in order to further demonstrate instances where OOD examples have low uncertainty, we now include **additional experiments for the rebuttal**, shown in lines 294-300 and Appendix A.2. Specifically, we benchmark 14 different models including ResNets, ViTs, and ConvNeXt V2s in the realistic setting where ImageNet-1K is ID, and we evaluate the performance of MSP for three different OOD datasets: Textures, iNaturalist, and ImageNet-OOD. We record the FPR@95, which indicates how many OOD examples are incorrectly classified as ID due to their low uncertainty (false positive), at a threshold where 95% of ID examples are correctly classified (true positive). The average FPR@95 over all models on all three OOD datasets is 66.5%; thus well over half of OOD examples are classified as ID due to their low uncertainty, and other methods such as max logit, energy score, and entropy all have similar FPR@95s over 60%. These experiment results demonstrate that OOD inputs having low uncertainty is a common pathology across many models and datasets.
>
> Please let us know if you have any further questions or comments, and we look forward to your response.

---

> ### Comment · Area_Chair_nFhh · 2024-12-02
>
> Dear Reviewer K5mb,
>
> This paper has polarized opinions among reviewers. The authors have submitted their rebuttal. Could you please take a moment to review their response? Thank you for your efforts and contributions to ICLR'25.
>
> Best regards,
>
> Your Area Chair

---

> ### Comment · Reviewer_K5mb · 2024-12-02
>
> I appreciate the authors' responses. I do not believe there is a significant misunderstanding. Major claims proposed in this work are intuitive and already recognized in the literature.
>
> The statement that "prior works are not focused on explaining why a paradigm for OOD detection is fundamentally flawed" appears to be incorrect. This problem has been well-established and rigorously explored. I recommend authors review [1] and many of its follow-up works. Unfortunately, the paper does not mention or discuss this line of research. Specifically, prior works have explored multiple conditions for OOD detection to be successful (e.g., conditions on hypothesis space and ID-OOD separability).
>
> [1] Fang et al., Is Out-of-Distribution Detection Learnable?, NeurIPS 2022 (Outstanding Paper)

---

> > ### Author Response · Authors · 2024-12-03
> >
> > Thank you for the reference. “Is OOD Detection Learnable?” provides many valuable insights; however, its focus on the feasibility of OOD detection across *any* learning algorithm is very different from our work. Rather than reducing OOD detection to the binary labels of “learnable” and “unlearnable” (under which all near-OOD settings arguably fall), our work explicitly identifies specific pathologies which arise from practical settings, and fills a critical gap in the literature.
> >
> > For instance, Fang et al. claim that “OOD detection is learnable in image-based scenarios when ID images have clearly different semantic labels and styles from OOD images (far-OOD).” However, their focus on theoretical learnability leaves important open questions regarding practical implementation: learnability does not imply a successful OOD detection method is easily found, since learnability is defined over all possible OOD detection functions rather than the ones achieved through standard training techniques. In fact, our paper provides examples of far-OOD detection where standard methods like MSP fail, indicating that learnability is insufficient for practical success. Moreover, Fang et al. prove that OOD detection is not learnable when the ID and OOD data have overlap (Theorem 4). This overlap is regularly the case for near-OOD detection, and the authors actually note that “developing a theory to understand the feasibility of near-OOD detection is still an open question.” Our paper aims to address this problem with new insights into near-OOD detection that are distinct from prior works.
> >
> > Specifically, our paper directly evaluates the performance of OOD detection methods in real-world settings, bridging the gap between theory and practice by showcasing the practical limitations of popular OOD detection methods across modern architectures and pre-training methods. We identify specific pathologies, such as the existence of irrelevant features for OOD detection, and we are first to explicitly quantify the extent to which these pathologies exist in modern settings with large models trained on internet-scale data. We also identify the scaling trends of these behaviors to obtain a comprehensive understanding. In contrast, Fang et al. focus on theoretical approaches and only perform one experiment using a four-layer neural network on a classification problem for two-dimensional data.
> >
> > Furthermore, it is not evident that our identified pathologies are well-known. In fact, many papers published after Fang et al. continue to propose new feature or logit-based methods, or support their use [2, 3, 4]. For example, the ImageNet-OOD paper [5] advocates for the performance of MSP, noting its consistency across many datasets. In contrast, our paper presents clear evidence that methods like MSP have fundamental pathologies, and we clearly exemplify their limitations on real-world datasets.
> >
> > [2] T2FNorm: Train-time Feature Normalization for OOD Detection in Image Classification, 2024\
> > [3] Learning with Mixture of Prototypes for Out-of-Distribution Detection, 2024\
> > [4] ReweightOOD: Loss Reweighting for Distance-based OOD Detection, 2024\
> > [5] ImageNet-OOD: Deciphering Modern Out-of-Distribution Detection Algorithms, 2023
> >
> > We are happy to add this discussion to our paper, and we think that the referenced work is complementary to our perspective. However, our novel identification of real-world pathologies and thorough empirical analysis differentiate us from previous work.

---

> ### Comment · Reviewer_K5mb · 2024-12-03
>
> I appreciate the authors for the detailed response and clarifications.
>
> However, I am still not convinced that the phenomena described in the paper constitute novel "pathologies" rather than limitations already evident in the literature. The main reason I mention the reference work on ID-OOD learnability is that a study of learnability presents valuable insights for the research community than what is currently provided in the paper.
>
> As most practical OOD detection methods are heuristics-based (e.g., logit or feature-based), the failure modes stated in the paper are well expected. For example, regarding feature-based OOD detection: "If the OOD and ID features are indistinguishable, then no feature-based methods can perform well." Given that no OOD detection methods achieve perfect performance, for any pre-trained model, one can easily find cases where (near or far) OOD samples overlap with ID in terms of learned features.
>
> While the authors claim this is one mode of pathology, I view this as a natural consequence of the heuristic approaches (among various other factors) being used in practice.  A positioning paper would benefit from studying the problem more rigorously, beyond a handful of case studies. There are numerous factors in practice, including training data, loss functions, optimization algorithm, and model architectures, that can influence these behaviors and warrant a rigorous examination.

---

> > ### Author Response · Authors · 2024-12-03
> >
> > Respectfully, we strongly disagree with your assessment. As we have discussed, there are many prominent recent papers that embrace and further extend the families of OOD detection procedures that we show have fundamental pathologies, clearly demonstrating that these issues are not in fact well-known. We have also explained in detail how the paper you referenced does not significantly overlap with our work, despite your claim that it indicates that the issues we identify are well-known.
> >
> > Furthermore, although there may be some hypotheses regarding the failure modes of these methods, prior works do not generally show when these pathologies occur. For instance, we demonstrate that there are many cases where the features are almost fully distinguishable (as determined by the high oracle classification accuracy in Figure 4) but feature-based OOD detection methods still fail. This directly contradicts your claim that “for any pre-trained model, one can easily find cases where (near or far) OOD samples overlap with ID in terms of learned features.” Our paper provides the necessary clarification and correction to commonly-held but inaccurate beliefs.
> >
> > Earlier, you had critiqued our work based on incorrect assertions, for example claiming that our conclusions are "drawn from experiments involving only a single model and dataset.” However, we have already provided a thorough analysis of hundreds of combinations of models and datasets, using over 50 different models on a wide variety of pre-training datasets, and we have added 168 new experimental results in the rebuttal alone. While it is always possible to request additional experiments on even more model architectures, pre-training datasets, etc, our paper is relatively exhaustive and contains comprehensive empirical results. In contrast, the paper you reference, while valuable, contains only a single toy experiment involving 2-D datapoints, which does not reflect any real-world OOD detection problems. Furthermore, our paper also includes a section dedicated to analyzing the impact of pre-training and model architecture involving modern state-of-the-art approaches, a section which you explicitly stated “would provide value”. We would also like to reiterate that our conclusions are foundational: while they are clearly exemplified by the empirical evidence, they are grounded in conceptual limitations.

---

> > > ### Comment · Reviewer_K5mb · 2024-12-03
> > >
> > > I appreciate the authors' response, but I feel concerned that my critiques have been mischaracterized as incorrect assertions. My core questions are ignored rather than addressed substantively. I believe a more constructive dialogue would better serve to improve the manuscript's quality.
> > >
> > > Let me clarify my position by revisiting my original critique about "Lack of Scientific Rigor":
> > >
> > > - "Lack of Scientific Rigor: The paper lacks scientific rigor in most of its experimental sections. Each section follows a similar format where a failure mode is claimed (often an obvious one) and then demonstrated with a basic experiment (e.g., “To demonstrate feature overlap, we train a ResNet-18 on a subset of CIFAR-10 classes,” L247). These conclusions are drawn from experiments involving only a single model and dataset, leaving open questions about the generalizability of the findings across different training and evaluation settings. "
> > >
> > > Even in the revised version, the examples remain limited:
> > >  -  "To demonstrate feature overlap, we train a ResNet-18 on a subset of CIFAR-10 classes: airplane, cats, and trucks." (L241-L242)
> > > -  Even when using a ResNet-50 trained on ImageNet-1K, Figure A.1 (right).
> > >
> > > To clearly illustrate my point, in the revised version, when discussing generative models (Sec 5.6), the examples appear arbitrary and methodologically inconsistent:
> > >
> > > - a toy example of Gaussian distribution "where the ID data is drawn from N (0, 1) and the OOD data is drawn from N (2, 1)".
> > > - RealNVP trained on CelebA (ID) vs CIFAR-10 (OOD)
> > > - Gaussian Mixture Model (GMM) based on features of ResNet-50 pre-trained on ImageNet-1k (ID) vs. ImageNet-OOD (OOD)
> > > -  DiT trained on ImageNet-1k (ID) vs. DTD (OOD)
> > >
> > > These examples lack scientific rigor in several ways:
> > > - There's no consistent choice of ID/OOD datasets across different models
> > > - Each model is tested with a single ID/OOD pair
> > > - The choices of dataset pairs seem arbitrary without justification
> > > - There's no systematic control of variables to isolate the effects being studied
> > >
> > > For a paper claiming to identify fundamental pathologies in generative OOD detection, one would expect:
> > > - A unified choice of ID/OOD datasets across all models
> > > - Multiple ID/OOD pairs for each model to demonstrate consistency
> > > - Controlled experiments that systematically vary one factor while keeping others constant
> > >
> > > I want to emphasize: the total number of models evaluated is not my primary concern. To repeat, my judgements are the following:
> > >
> > > "As most practical OOD detection methods are heuristics-based (e.g., logit or feature-based), the failure modes stated in the paper are well expected. For example, regarding feature-based OOD detection: "If the OOD and ID features are indistinguishable, then no feature-based methods can perform well." Given that no OOD detection methods achieve perfect performance, for any pre-trained model, one can easily find cases where (near or far) OOD samples overlap with ID in terms of learned features.
> > >
> > > While the authors claim this is one mode of pathology, I view this as a natural consequence of the heuristic approaches (among various other factors) being used in practice. A positioning paper would benefit from studying the problem more rigorously, beyond a handful of case studies. There are numerous factors in practice, including training data, loss functions, optimization algorithm, and model architectures, that can influence these behaviors and warrant a rigorous examination."
> > >
> > > There exists a lack of convincing evidence for individual claims made by the paper. I do not see an improvement of scientific rigor in the revised manuscript.

---

> > > > ### Author Response · Authors · 2024-12-04
> > > >
> > > > We are not misconstruing your critiques but instead pointing out legitimate factual errors. For example, your initial review says our conclusions are drawn from only a single model and dataset. Both our initial and revised submission in fact had many demonstrations: for example, for the pathologies of feature-based methods, we trained four different models: ResNet-18, ResNet-50, ViT-S/16, and ViT-B/16, and used three different OOD datasets: ImageNet-OOD, Textures, and iNaturalist, as described in L202-L205. This is significantly more experiments than you acknowledge in your latest response.
> > > >
> > > > Furthermore, your claim that each model is tested with a single ID/OOD pair is also inaccurate. As we previously stated, our experiments primarily use ID datasets of ImageNet-1k with OOD datasets of ImageNet-OOD, Textures, and iNaturalist. In our rebuttal alone, we see in Table A.1 that we have benchmarked fourteen models on the same three ID/OOD combinations, and we consistently show similar trends throughout. Moreover, we have many experiments which highlight how performance changes as we vary specific factors. For example, Figure 4 shows the impact of ID data accuracy on the various pathologies we identify. Our rebuttals to your claims are not based on subjective opinion but rather on objective facts, and we are confused about why you are not willing to acknowledge the discrepancies.
> > > >
> > > > Unlike the factual errors in your review, the perceived “level of rigor” of course is subjective. But we must again disagree with your perception. Relative to standard evaluations, we have exhaustively considered the behaviors of a diverse set of models over many ID/OOD settings. You also highlighted, on the last day of the reviewer-author discussion period, Feng et al. [1] as a reason that what we argue is well-known, but we explained in detail why our arguments and contributions importantly differ from [1]. Furthermore, our paper explicitly refutes claims you make in your review, such as “for any pre-trained model, one can easily find cases where (near or far) OOD samples overlap with ID in terms of learned features”, which in fact highlights the novelty and importance of our work: the results appear to contradict your prior beliefs.
> > > >
> > > > Finally, we have now several times explained how our arguments are largely not arguments by example but instead conceptual limitations. We have empirically demonstrated these limitations across diverse real-world settings, but the core arguments are foundational.

---

### Author Response · Authors · 2024-11-14
**General Response**

We would like to thank all of the reviewers for their feedback. We present a general response to the reviewers here, and we also address individual reviewers in separate posts below.

OOD detection has been a vital component of AI safety and robustness, and many methods have been developed to address this problem. In our work, we are the first to highlight the fundamental pathologies of entire families of popular approaches to OOD detection, such as methods which utilize the features or logits of supervised models. We demonstrate that these approaches do not directly address OOD detection but instead answer entirely different questions, and we provide concrete failure modes across a diverse set of models and datasets. We do not propose yet another method which claims to improve OOD performance while following fundamentally pathological paradigms; instead, our research contribution lies in our novel demonstrations of the irresolvable failure modes when using supervised models trained on in-distribution data to detect out-of-distribution data.

Because there are many works which criticize specific OOD detection procedures, it may seem, superficially, as if our work lacks novelty. However, many SoTA OOD detection methods focus on adaptations within these existing families, often implying the opposing stance that improved methodology can address these innate limitations. For example, Sun et al [1] attribute the problem with feature-based approaches to the distributional parameterization and thus propose a feature-based approach. Wei et al [2] attribute the problem with logit-based approaches to model overconfidence and thus propose a novel logit-based approach. Wang et al [3] state that the performances of logit-based and feature-based methods “are limited by the singleness of their information source” and thus propose a novel hybrid approach. In contrast, we argue that even these novel methods, or any other logit-based or feature-based methods, still have fundamental limitations that arise from the misspecification of supervised training. No prior work has explored the inherent pathologies of OOD detection methods which cannot be mitigated by improved methodology.

We would also like to clarify that we have extensive experiments over a wide variety of models and datasets, and our results are not limited to toy examples. In fact, our paper includes empirical results for hundreds of combinations of model architectures, training procedures, and ID and OOD datasets.
- To demonstrate the pathologies of feature-based methods (section 4.1), we provide experiment results using four different models: ResNet-18, ResNet-50, ViT-S/16, and ViT-B/16. We use ImageNet-1k as our ID data, and we test on three different OOD datasets: ImageNet-OOD, Textures, and iNaturalist. We also use a ResNet-18 trained on CIFAR-10 for visualization purposes, since smaller feature dimensions and fewer classes lead to clearer 2D plots; furthermore, we also visualize failure modes for ResNet-50 on ImageNet-1k.
- To understand how these feature-based pathologies scale with model and dataset size (Section 5.1), we benchmark 12 different models of varying sizes and training methods, as listed in Appendix B.3.
- For pathologies of logit-based methods (Section 4.2), we show results for ResNet-18, ResNet-50, ViT-S/16, ViT-B/16, trained on ImageNet, and ViT-G/14 trained with DINOv2. We also use a LeNet-5 trained on CIFAR-10 for visualization.
- To understand the scaling behaviors of logit-based methods, we evaluate 54 models from timm and torchvision, with 9 different architectures and 6 different pre-training setups, listed in Appendix B.2.

While these experimental setups were included in the original paper, we have revised the text to highlight the diverse set of models and datasets benchmarked.

Even though it has become popular to use out-of-the-box supervised predictive models for OOD detection, we make the case that this family of approaches is fundamentally misspecified for this purpose. Due to the profound significance of this undertaking and the broad relevance of our work, we believe our paper provides an urgent and needed contribution to the community. We hope these points, and our responses, can be accounted for in the final assessment, and that reviewers may be open minded about revising their understanding.

[1] Sun et al, Out-of-Distribution Detection with Deep Nearest Neighbors\
[2] Wei et al, Mitigating Neural Network Overconfidence with Logit Normalization\
[3] Wang et al, ViM: Out-Of-Distribution with Virtual-logit Matching\

---

> ### Author Response · Authors · 2024-11-20
> **General Response Update**
>
> In response to reviewer feedback, we now include **additional experiments for the rebuttal** in order to further demonstrate instances where OOD examples have low uncertainty. These updates are reflected in lines 294-300 and Appendix A.2. Specifically, we benchmark 14 different models including ResNets, ViTs, and ConvNeXt V2 on the realistic setting where ImageNet-1K is ID, and we evaluate the performance of MSP for three different OOD datasets: Textures, iNaturalist, and ImageNet-OOD. We record the FPR@95, which indicates how many OOD examples are incorrectly classified as ID due to their low uncertainty (false positive), at a threshold where 95% of ID examples are correctly classified (true positive). The average FPR@95 over all models on all three OOD datasets is 66.5%; thus well over half of OOD examples are classified as ID due to their low uncertainty, and other methods such as max logit, energy score, and entropy all have similar FPR@95s over 60%. These experiment results demonstrate that OOD inputs having low uncertainty is a common pathology across many models and datasets.
>
> Please let us know if you have any further questions or comments!

---

### Comment · Area_Chair_nFhh · 2024-11-24

Dear Reviewers,

The public discussion phase is ending soon, and active participation is highly appreciated and recommended. Thanks for your efforts and contributions.

Best regards,

Your Area Chair

---

### Author Response · Authors · 2024-12-02
**General Response Update**

Numerous OOD detection methods have emerged in recent years, each identifying specific weaknesses of feature- or logit-based approaches and proposing tailored workarounds that still rely on these representations. In contrast, our work takes a fundamentally different perspective: instead of pursuing methodological advances designed to address minor and narrowly focused limitations, we critically examine the foundational limits of *any* feature-based approach. We explicitly quantify these limitations across an extensive range of models and datasets, and we identify that there are clear upper bounds for any feature or logit-based methods. Furthermore, our paper provides evidence that scaling up these models is not sufficient to solve these pathologies, and fundamentally different methods are required. Our paper provides a comprehensive understanding of the pathologies of OOD detection based on supervised models, offering valuable insights beyond the critiques of individual methods.

We closely examine the impacts of training details such as model architecture, pre-training method, OOD dataset, and more, systematically exploring how these factors influence the limitations of OOD detection methods. We show, for example, how extensive pre-training can soften some of the misspecifications we observe.

However, fully addressing all of the identified limitations significantly exceeds the scope of a typical research paper, which often identifies a single minor limitation that is narrow in scope as motivation for a methodological remedy (e.g., max-softmax -> max-logit). By contrast, resolving the pathologies we identify would require a fundamental paradigm shift within the field of OOD detection, similar in scope to resolving adversarial robustness or other well-established fields. Furthermore, the current state of OOD detection reflects years of contributions from the community, making it unrealistic to expect a single section of our paper to offer a resolution to these deeply-seated far-reaching challenges. Indeed, part of the point of the paper is to help constructively change the trajectory of this research area.

Although the contributions of a paper which proposes novel methodology are straightforward to identify, there is also significant value in contributions which advance our understanding of existing methods and prompt further research. There have been many successful works published at top venues which focus on exploration [1, 2, 3, 4, 5, 6]. For instance, in “How Good is the Bayes Posterior in Deep Neural Networks Really?” (ICML 2020, 373 citations) [7], the authors state in the conclusion that “Our work has raised the question of cold posteriors but we did not fully resolve nor fix the cause for the cold posterior phenomenon.” In “Understanding deep learning requires rethinking generalization” (ICLR 2016 best paper) [8], the authors conclude that “we have yet to discover a precise formal measure under which these enormous models are simple.” Had methodological innovation been required of these papers, they all would have been rejected. Our work follows a similar trajectory, where we focus on explicitly identifying the behaviors and pathologies of existing methods.

[1] Wilson et al, “The marginal value of adaptive gradient methods in machine learning” (NeurIPS 2017, 1278 citations)\
[2] Recht et al, “Do ImageNet Classifiers Generalize to ImageNet?” (ICML 2019, 1889 citations)\
[3] Lotfi et al, “Bayesian Model Selection, the Marginal Likelihood, and Generalization” (ICML 2022, Best Paper)\
[4] Bae et al, “If Influence Functions are the Answer, Then What is the Question?” (NeurIPS 2022)\
[5] Ilyas et al, “Adversarial Examples Are Not Bugs, They Are Features” (NeurIPS 2019, 2114 citations)\
[6] Rylan et al. "Are emergent abilities of large language models a mirage?." (NeurIPS 2023, Outstanding Paper). \
[7] Wenzel et al, “How Good is the Bayes Posterior in Deep Neural Networks Really?” (ICML 2020, 373 citations)\
[8] Zhang et al, “Understanding deep learning requires rethinking generalization” (ICLR 2016, Best Paper)

Our work fundamentally challenges the premise underlying many OOD detection methods through extensive conceptual and empirical demonstrations. We believe that it is timely and will be of great interest to the community, given the plethora of new OOD methods. We focus on a fundamental conceptual understanding of deeply-seated limitations, rather than methodologies intended to address minor and more narrowly focused limitations.

---

### Meta-Review · Area_Chair_nFhh · 2024-12-17

**Metareview:**

This paper tries to investigate the issues of existing OOD detection methods. The starting point of this article is quite high, and the authors claim that their findings are important and make a foundational contribution. However, one reviewer raised severe concerns about overclaim and the lack of scientific rigor, including inconsistent choice of datasets, no systematic control of variables, and arbitrary dataset choices without justification. Other reviewers also felt the research contribution of this paper is not clear. AC oversaw the whole review process. From my reading of this paper, I believe the critical analysis and identified failure mode of OOD detection methods are valuable and inspiring for researchers. However, many concerns raised by the reviewers are not minor and reasonable, which means this paper can not be published on ICLR in its current version.

**Additional Comments On Reviewer Discussion:**

Reviewers raised many concerns about the rigorous experimental design and contribution. The authors' rebuttal did not address the majority of these concerns. At the end of the discussion, one reviewer strongly recommended rejection, and three of the four reviewers pushed for rejection, putting this paper below the acceptance bar.

---

### Decision · Program_Chairs · 2025-01-22

Reject